

# DMS oxidation and sulfur aerosol formation in the marine troposphere: a focus on reactive halogen and multiphase chemistry

Qianjie Chen[1*], Tomás Sherwen[2], Mathew Evans[2,3], and Becky Alexander[1]

[1]Department of Atmospheric Sciences, University of Washington, Seattle, WA, USA
[2]Wolfson Atmospheric Chemistry Laboratories, Department of Chemistry, University of York, York, UK
[3]National Centre for Atmospheric Science (NCAS), University of York, York, UK
[*]*Now at* Department of Chemistry, University of Michigan, Ann Arbor, MI, USA

*Correspondence to*: Becky Alexander (beckya@atmos.washington.edu)

**Abstract.** The oxidation of dimethyl sulfide (DMS) in the troposphere and subsequent chemical conversion into sulfur dioxide ($SO_2$) and methane sulfonic acid (MSA) are key processes for the formation and growth of sulfur-containing aerosol and cloud condensation nuclei (CCN), but is highly simplified in large-scale models of the atmosphere. In this study, we implement a series of gas-phase and multiphase sulfur oxidation mechanisms into the GEOS-Chem global chemical transport model, including two important intermediates dimethyl sulfoxide (DMSO) and methane sulphinic acid (MSIA), to investigate the sulfur cycle in the global marine troposphere. We found that DMS is mainly oxidized in the gas phase by OH (66%), $NO_3$ (16%) and BrO (12%) globally. DMS+BrO is important for the model's ability to reproduce the observed seasonality of surface DMS mixing ratio in the Southern Hemisphere. MSA is mainly produced from multiphase oxidation of MSIA by $O_{3(aq)}$ (69%) and $OH_{(aq)}$ (25%) in cloud droplets and aerosols. Aqueous-phase reaction with OH accounts for 38% of MSA removal globally and is important for the model's ability to reproduce observations of MSA/nss$SO_4^{2-}$ ratio in the Southern Hemisphere. The modeled conversion yield of DMS into $SO_2$ and MSA is 78% and 13%, respectively, compared to 91% and 9% in the standard model run that includes only gas-phase oxidation of DMS by OH and $NO_3$. The largest uncertainties for modeling sulfur chemistry in the marine boundary layer (MBL) are unknown concentrations of reactive halogens (BrO and Cl) and $OH_{(aq)}$ concentrations in cloud droplets and aerosols. To reduce uncertainties in MBL sulfur chemistry, we should prioritize observations of reactive halogens and $OH_{(aq)}$.

## 1 Introduction

The biogenic emission of dimethyl sulfide (DMS: $CH_3SCH_3$) from the ocean is the largest natural sulfur source to the atmosphere (Andreae, 1990). After emission, DMS is mainly oxidized in the troposphere, with a lifetime against oxidation of 1-2 days (Chin et al., 1996; Boucher et al., 2003; Breider et al., 2010). The oxidation of DMS and subsequent formation of other sulfur species such as sulfuric acid ($H_2SO_4$) and methane sulfonic acid (MSA: $CH_3SO_3H$) are crucial for the formation and evolution of natural aerosols and clouds in the marine boundary layer (MBL) and thus have profound climate implications (Charlson et al., 1987; von Glasow and Crutzen, 2004; Thomas et al., 2010). In particular, Carslaw et al. (2013)



pointed out that natural aerosols such as those that originate from DMS oxidation account for the largest uncertainty of aerosol radiative forcing in climate models.

The atmospheric fate of DMS determines the extent to which DMS affects our climate system. Production of $H_2SO_4$ and
MSA from gas-phase oxidation of DMS-derived products can result in nucleation of new particles under favorable conditions (Kulmala et al., 2000; Chen et al., 2015), with implications for aerosol and CCN number concentrations. Sulfate and MSA formed in the aqueous phase will not result in new particle formation, but will impact the aerosol size distribution with implications for cloud microphysical properties (Kreidenweis and Seinfeld, 1988; Kaufman and Tanre, 1994). The oxidation mechanisms of DMS and subsequent formation of sulfate and MSA are, however, very complicated and still not
well understood even after decades of research (Ravishankara et al., 1997; Barnes et al., 2006; Hoffmann et al., 2016). Large-scale models of atmospheric chemistry typically contain very simplified DMS chemistry, and often ignore potentially important reaction intermediates. Most of these models include oxidation of DMS by OH and $NO_3$ radicals, directly producing $SO_2$ and MSA, and ignore the formation of dimethyl sulfoxide (DMSO: $CH_3SOCH_3$) and methane sulphinic acid (MSIA: $CH_3SO_2H$) intermediates (Chin et al., 1996; 2000; Gondwe et al., 2003; 2004; Berglen et al., 2004; Kloster et al.,
2006). Nevertheless, previous large-scale modeling studies suggested that BrO could be an important sink for DMS globally (up to 30%), especially in the remote MBL where BrO mixing ratios can reach ppt levels (Boucher et al., 2003; von Glasow et al., 2004; Breider et al., 2010; Khan et al., 2016). Other oxidants that may be important for DMS oxidation include Cl radicals in the gas phase (von Glasow and Crutzen, 2004; Hoffmann et al., 2016) and $O_3$ in the gas and aqueous phase (Boucher et al., 2003; Hoffmann et al., 2016).

Some large-scale models have simulated the formation of the DMSO intermediate from DMS oxidation (Pham et al., 1995; Cosme et al., 2002; von Glasow et al., 2004; Castebrunet et al., 2009), which is important as DMSO is highly water soluble (Henry's law constant ($H_{DMSO}$) on the order of $10^7$ M atm$^{-1}$) and can undergo dry and wet deposition in addition to gas- and aqueous-phase oxidation to MSA or $SO_2$ (Lee and Zhou, 1994; Campolongo et al., 1999; Barnes et al., 2006; Zhu et al.,
2006; Hoffmann et al., 2016). In the cloud-free MBL, DMSO is mainly produced by DMS+BrO and DMS+ $OH_{(g)}$ via the addition channel and is oxidized by OH in the gas phase. In the cloudy MBL, DMSO is mainly produced via DMS+$O_{3(aq)}$ and oxidized via DMSO+$OH_{(aq)}$ in the aqueous phase (Hoffmann et al., 2016). Knowledge about aqueous-phase concentrations of OH in cloud droplets and aerosols is still very limited. Modeled $OH_{(aq)}$ concentrations are on the order of $10^{-14}$-$10^{-12}$ M (Jacob, 1986; Matthijsen et al., 1995; Jacob et al., 1989; Herrmann et al., 2000). However, recent observations of $OH_{(aq)}$,
which are derived from the concentrations of dissolved organic compound, are about two orders of magnitude lower ($10^{-16}$-$10^{-14}$ M) (Arakaki et al., 2013; Kaur and Anastasio, 2017). In addition to aqueous-phase oxidation of DMSO by $OH_{(aq)}$, a box modeling study by Zhu et al. (2006) suggested that $SO_4^-$ and $Cl_2^-$ could contribute to 34% and 10% of DMSO oxidation in the aqueous phase, respectively, with $SO_4^-$ and $Cl_2^-$ concentrations of $1\times10^{-12}$ M and $1\times10^{-11}$ M (Herrmann et al., 2000), respectively. It should be noted that $OH_{(aq)}$, $SO_4^-$ and $Cl_2^-$ concentrations are poorly known and the contribution of these




species to DMSO oxidation will depend on their concentrations.

MSIA is generally not included in large-scale models, though it has been considered in some one-dimensional or box models (Lucas and Prinn, 2002; von Glasow and Crutzen, 2004; Zhu et al., 2006; Hoffmann et al., 2016). The Henry's law constant

of MSIA has not been measured directly but is thought to be larger than that of DMSO and smaller than that of MSA, on the order of $10^8$ M atm$^{-1}$ (Barnes et al., 2006). MSIA is mainly produced from oxidation of DMSO by OH in both the gas and aqueous phase, and removed via further oxidation by OH and $O_3$ in both the gas and aqueous phase and $Cl_2^-$ in the aqueous phase (von Glasow and Crutzen, 2004; Zhu et al., 2006; Barnes et al., 2006; Hoffmann et al., 2016). Only oxidation of MSIA by OH in the gas phase produces $SO_2$, all other pathways lead to MSA formation. The contribution of each pathway towards

MSIA oxidation depends on the concentration of each oxidant. Zhu et al. (2006) suggested $Cl_2^-$ is more important than $OH_{(aq)}$ for MSIA oxidation in the aqueous phase when assuming a $Cl_2^-$ concentration of $1 \times 10^{-11}$M (Herrmann et al., 2000), while Hoffmann et al. (2016) suggested the opposite with a lower $Cl_2^-$ concentration ($1.5 \times 10^{-12}$M).

The only source of MSA in the marine troposphere is from oxidation of DMS emitted from the marine biosphere. It thus

contains information on both DMS emission flux and chemistry. It has been proposed as an ice-core proxy for sea ice extent in past climates, as a result of melting sea ice releasing nutrients to stimulate phytoplankton growth to produce DMS (Curran et al., 2003; Abram et al., 2010). Other factors such as oxidation mechanisms of DMS and atmospheric circulation can also affect MSA abundance in ice core records (Becagli et al., 2009; Hezel et al., 2011). As DMS is the dominant sulfur source of both MSA and non-sea-salt sulfate (nss$SO_4^{2-}$) in the remote marine troposphere, the MSA/nss$SO_4^{2-}$ molar ratio there reflects

sulfur chemistry. In addition, the MSA/nss$SO_4^{2-}$ molar ratio has often been used as a measure of marine biogenic contribution to total atmospheric sulfate formation, as nss$SO_4^{2-}$ has both anthropogenic and natural origins while MSA has only a natural source (Andreae et al., 1999; Savoie et al., 2002; Gondwe et al., 2004). MSA is very water soluble, with a Henry's law constant on the order of $10^9$ M atm$^{-1}$ (Campolongo et al., 1999), and is mainly removed from the atmosphere via wet and dry deposition with a lifetime of about a week (Pham et al, 1995; Chin et al., 1996; 2000; Cosme et al., 2002; Hezel

et al., 2011). One-dimensional modeling studies by Zhu et al. (2006) and von Glasow and Crutzen (2004) suggested that the oxidation of MSA by $OH_{(aq)}$ in the aqueous phase to form $SO_4^{2-}$ in the MBL could also be a significant loss process of MSA (3-27%) (Zhu et al., 2006; von Glasow and Crutzen, 2004), while a box modeling study by Hoffmann et al. (2016) found it negligible (2%). The different conclusions regarding the role of reaction of MSA with $OH_{(aq)}$ is due to different assumptions regarding $OH_{(aq)}$ concentrations, which are highly uncertain.

In this study, we expand upon the current simplified DMS chemistry in a global chemical transport model GEOS-Chem, including the DMSO and MSIA intermediates. We investigate the role of gas-phase and multiphase oxidation of DMS, DMSO, MSIA and MSA for determining their spatial distribution, seasonality, and lifetime and the implications for the MBL and global sulfur budget. Observations of DMS mixing ratio from 4 locations and MSA/nss$SO_4^{2-}$ ratio from 23





locations around the globe obtained from previous studies are used to assess the model. We conclude with recommendations for future laboratory experiments and field campaigns, and recommendations for sulfur chemistry that should be included in large-scale models of atmospheric chemistry and climate.

## 2 GEOS-Chem model

In this study, we use a global 3-D chemical transport model GEOS-Chem v9-02 (http://www.geos-chem.org/), which is driven by assimilated meteorological data from the NASA Goddard Earth Observing System (GEOS-5, http://gmao.gsfc.nasa.gov). It contains detailed $HO_x$-$NO_x$-VOC-ozone-$BrO_x$ tropospheric chemistry originally described in Bey et al. (2001), with updated $BrO_x$ and sulfate chemistry described in Parrella et al. (2011), Schmidt et al. (2016) and Chen et al. (2017). The sulfate-nitrate-ammonium aerosol simulation is fully coupled to gas-phase chemistry (Park et al., 2004),

with aerosol thermodynamics described in Pye et al. (2009). The sea salt aerosol simulation is described in Jaeglé et al. (2011) and bulk cloud water pH is calculated as described in Alexander et al. (2012). The model contains detailed deposition schemes for both gas species and aerosols (Liu et al., 2001; Amos et al., 2012; Zhang et al., 2001; Wang et al., 1998). All simulations are performed at 4°×5° horizontal resolution and 47 vertical levels up to 0.01 hPa after a model spin up of one year. Year 2007 is chosen as a reference year to be consistent with Schmidt et al. (2016) and Chen et al. (2017). DMS

emission flux from the ocean ($F$) is parameterized following Lana et al. (2011): $F = k_T C_w$, where gas transfer velocity $k_T$ is a function of sea surface temperature and wind speed and $C_w$ is the DMS concentrations in sea water obtained from Lana et al. (2011). In a sensitivity simulation, we used $C_w$ from Kettle et al. (1999).

The standard model contains only three DMS oxidation pathways in the original version, which produces $SO_2$ and MSA

directly ($R1$-$R3$), following Chin et al. (1996) with updated reaction rate coefficients from Burkholder et al. (2015):

$$DMS + OH \xrightarrow{abstraction} SO_2 + CH_3O_2 + CH_2O \tag{$R1$}$$

$$DMS + OH \xrightarrow{addition} 0.75SO_2 + 0.25MSA \tag{$R2$}$$

$$DMS + NO_3 \rightarrow SO_2 + HNO_3 + CH_3O_2 + CH_2O \tag{$R3$}$$

The yields of $SO_2$ and MSA for the addition channel of DMS+OH reaction are originally from Chatfield and Crutzen (1990), who made simplified assumptions in their 2-D model based on previous laboratory experiments and modeling studies. It should be noted that only gas-phase chemistry was considered when they made the assumptions of the yields of $SO_2$ and MSA, which might not represent the real atmosphere as multiphase chemistry has been suggested to be the biggest source of

MSA in the atmosphere (Zhu et al., 2006; Hoffmann et al., 2016).


We add the DMSO and MSIA intermediates as two new advected chemical tracers, which undergo chemical production and loss, transport and deposition in the model. We add 12 new chemical reactions in the model, including gas-phase oxidation of DMS by OH (addition channel, modified to produce DMSO instead of MSA), BrO, Cl and $O_3$, multiphase oxidation of DMS by $O_3$, both gas-phase and multiphase oxidation of DMSO by OH, both gas-phase and multiphase oxidation of MSIA

by OH and $O_3$, and multiphase oxidation of MSA by OH, as shown in Table 1. The rate coefficients for all gas-phase sulfur reactions are obtained from the most recent JPL report (Burkholder et al., 2015), except for MSIA+$O_{3(g)}$ ($k_{MSIA+O3(g)}$) (Lucas and Prinn, 2002; von Glasow and Crutzen, 2004). The sulfur product yields for gas-phase reactions are obtained from various laboratory and modeling studies as indicated in Table 1. All oxidants (OH, $O_3$, $H_2O_2$, BrO, HOBr) are simulated in the full chemistry scheme, except for Cl radicals. We used monthly mean Cl mixing ratios from Sherwen et al. (2016), which

considered Cl-Br-I coupling but did not include chlorine production from the HOBr+$Cl^-$+$H^+$ reaction on sea salt aerosols that was suggested to be the largest tropospheric chlorine source in Schmidt et al. (2016). We imposed a diurnal variation of Cl abundances based on solar zenith angle, similar to the offline simulation of OH abundances in GEOS-Chem (Fisher et al., 2017).

For the multiphase reactions DMS+ $O_{3(aq)}$, DMSO+ $OH_{(aq)}$, MSIA+ $OH_{(aq)}$, MSIA+ $O_{3(aq)}$ and MSA+ $OH_{(aq)}$ in cloud droplets and aerosols, we assume a first-order loss of the sulfur species, following the parameterization described in Ammann et al. (2013) and Chen et al. (2017):

$$\frac{d[X]}{dt} = -\frac{c\gamma}{4}A[X] , \qquad (E4)$$

where X represents DMS, DMSO, MSIA or MSA; c is the average thermal velocity of X; A is the total surface area
concentration of aerosols or cloud droplets; $\gamma$ is the reactive uptake coefficient of X that involves gas diffusion ($\gamma_d$), mass accommodation ($\alpha_b$) and chemical reaction ($\Gamma_b$) in the aerosols or cloud droplets, as calculated in $E5$-$E7$.

$$\frac{1}{\gamma} = \frac{1}{\gamma_d} + \frac{1}{\alpha_b} + \frac{1}{\Gamma_b} \qquad (E5)$$

$$\gamma_d = \frac{4D_g}{cr} \qquad (E6)$$

$$\Gamma_b = \frac{4H_X RT \sqrt{D_{l,X} k_{X+Y}[Y]} f_r}{c} \qquad (E7)$$

where $r$ is radius for aerosols or cloud droplets ($\mu$m); $D_g$ is the gas phase diffusion coefficient of X ($m^2 s^{-1}$), calculated as a function of air temperature and air density following Chen et al. (2017). $H_X$ and $D_l$ are the Henry's law constant (M atm$^{-1}$) and liquid phase diffusion coefficient ($m^2 s^{-1}$) of X, which are summarized in Table 2; $R$ (=8.31×10$^{-2}$ L bar mol$^{-1}$ K$^{-1}$) is the universal gas constant. $T$ is air temperature (K); [Y] (= $[OH_{(aq)}]$ or $[O_{3(aq)}]$) is the aqueous phase concentration of the oxidant in aerosols or cloud droplets (M), where $[O_{3(aq)}]$ is calculated assuming gas-liquid equilibrium and $[OH_{(aq)}]$ is calculated
following Jacob (2005) ($[OH_{(aq)}]=\beta[OH_{(g)}]$, $\beta$=1×10$^{-19}$ M cm$^3$ molecule$^{-1}$). This is about two orders of magnitude higher than




[OH$_{(aq)}$] calculated indirectly from dissolved organic compound observations in Arakaki et al. (2013) and Kaur and Anastasio (2017). Thus, we conduct a sensitivity simulation reducing [OH$_{(aq)}$] by two orders of magnitude (Table 3). $k_{X+Y}$ is the aqueous-phase reaction rate coefficient between X and Y (M$^{-1}$ s$^{-1}$), as summarized in Table 1. $f_r$ ($=\coth(r/l) - l/r$) is the reacto-diffusive correction term, which compares the radius of aerosols or cloud droplets ($r$) with the reacto-diffusive length

scale of the reaction ($l = \sqrt{D_l/(k_{X+Y}[Y])}$) (Ammann et al., 2013). The mass accommodation coefficients ($\alpha_b$) of DMS, DMSO, MSIA and MSA are given in Table 2. Multiphase sulfur reactions added in the model are only activated when air temperature is above 273 K, to be consistent with the temperature at which their rate constants were obtained (Table 1).

Ten model simulations were performed in order to investigate the importance of individual reactions for MBL sulfur
chemistry and are described in Table 3. These simulations were designed to explore the role of DMS chemistry versus emissions for the DMS budget, and the importance of gas-phase reactive halogen chemistry and multiphase chemistry for all sulfur-containing compounds.

## 3 Results and Discussion

### 3.1 DMS budget

Figure 1 shows the global sulfur budgets for the model run including DMSO and MSIA intermediates and all 12 new reactions ($R_{all}$). The DMS emission flux from the ocean to the atmosphere ($F_{DMS}$) is 22 Tg S yr$^{-1}$, which is similar to that (24 Tg S yr$^{-1}$) reported in Hezel et al. (2011) and within the range (11-28 Tg S yr$^{-1}$) reported in the literature (Spracklen et al., 2005 and reference therein). $F_{DMS}$ is 18 Tg S yr$^{-1}$ when using sea surface DMS concentrations from Kettle et al. (1999). The tropospheric burden of DMS is 75 Gg S, which is within the range of 20-150 Gg S reported in Faloona et al. (2009), and is
39% lower than the standard model run ($R_{std}$). The lifetime of DMS is 1.3 days in $R_{all}$, compared to 2.1 days in $R_{std}$. Surface DMS mixing ratios are highest over Southern Ocean ($\approx$400 ppt) (Fig. 2a) where DMS emissions are highest during summer (Lana et al., 2011) and DMS chemical destruction is small due to low OH abundance at high latitudes (DMS lifetime of 2-5 days over Southern Ocean). DMS mainly resides in the lower troposphere, with 86% of the tropospheric burden below 2 km. DMS is mainly oxidized in the gas phase by OH (37% via abstraction channel and 29% via addition channel), followed by
NO$_3$ (16%). The global contribution of OH and NO$_3$ to DMS oxidation from previous studies is 50%-70% and 20%-30%, respectively, depending mainly on which other oxidants are included (Boucher et al., 2003; Berglen et al., 2004; Breider et al., 2010; Khan et al., 2016). The oxidation of DMS by OH occurs mainly during daytime while oxidation by NO$_3$ occurs mainly at night due to low nighttime OH production and rapid photolysis of NO$_3$ during daytime. Fig. 3 shows the global, annual mean distribution of the fractional importance of different DMS oxidation pathways. The relative importance of OH
for the oxidation of DMS ($f_{[I]DMS+OH(g)}$) is typically greater than 50% over the oceans. The relative importance of NO$_3$ for the oxidation of DMS ($f_{[I]DMS+NO3}$) is typically low over the remote oceans (<10%), but high over the continents and coastal



regions (>40%) where NO$_x$ emissions are highest. It should be noted, however, that DMS abundance is low over continents (Fig. 2a).

The relative importance of BrO oxidation of DMS ($f_{[I]DMS-BrO}$) is 12% (global, annual mean), which is within the range

suggested by Khan et al. (2016) (8%) and Breider et al. (2010) (16%). $f_{[I]DMS-BrO}$ is highest (>30%) over the Southern Ocean and Antarctica, especially during winter, due to high BrO (up to 0.5 ppt) and low OH and NO$_3$ abundance. The main uncertainty of the importance of BrO for DMS oxidation resides in the tropospheric BrO abundance, which is rarely measured and is still not well quantified in global models (von Glasow et al., 2004; Simpson et al., 2015). The BrO in our model generally underestimates satellite observations, especially over mid- and high-latitudes (Chen et al., 2017), suggesting

that our modeled estimate of the importance of DMS+BrO may be biased low. In order to quantify the contribution of BrO to DMS oxidation, we need to better quantify the BrO abundance through both observation and model development.

The fractional contribution of Cl to DMS oxidation ($f_{[I]DMS+Cl}$) is 4% globally and generally less than 10% everywhere. In comparison, von Glasow and Crutzen (2004) calculate that about 8% of DMS is oxidized by Cl in the cloud-free MBL

during summer in a 1-D model. Hoffmann et al. (2016) estimated that about 18% of DMS is oxidized by Cl under typical MBL conditions in a box model. Both studies used the same $k_{DMS+Cl}$ as in our study, but Cl concentrations were not reported in either study. The annual-mean tropospheric Cl concentration used in this study is $1.1 \times 10^3$ atoms cm$^{-3}$, which is similar to that ($1.3 \times 10^3$ atoms cm$^{-3}$) in another recent 3-D modeling study (Hossaini et al., 2016). As suggested by Sherwen et al. (2016), Cl concentration could be underestimated in our study, due at least in part to the missing chlorine source from sea

salt aerosols and anthropogenic chloride emissions. The largest uncertainty for the importance of Cl for the oxidation of DMS resides in our limited knowledge of Cl concentrations in the troposphere. Due to the difficulty of directly observing Cl, estimates of its abundance are usually derived from non-methane hydrocarbons (NMHC) observations. Using this method, Cl concentration is estimated to be on the order of $10^4$ atoms cm$^{-3}$ ($0.2-80 \times 10^4$ atoms cm$^{-3}$) in the MBL and Antarctic boundary layer (Jobson et al., 1994; Singh et al., 1996; Wingenter et al., 1996; 2005; Boundries and Bottenheim, 2000;

Arsene et al., 2007; Read et al., 2007), with highest concentrations over Tropical Pacific during autumn (Singh et al., 1996). Another uncertainty in the atmospheric implications of DMS+Cl originates from its sulfur products, which are most likely CH$_3$SCH$_2$ via the abstraction channel and (CH$_3$)$_2$S-Cl adduct via the addition channel (Barnes et al., 2006). The CH$_3$SCH$_2$ will likely be further oxidized into SO$_2$, similar to the abstraction channel of DMS+OH, while the (CH$_3$)$_2$S-Cl adduct could react with O$_2$ to produce DMSO. Atkinson et al. (2004) estimated that 50% of DMS+Cl occurs through the abstraction

channel and 50% occurs through the addition channel at 298 K and 1 bar pressure, but the abstraction channel could account for more than 95% at low pressure (Butkovskaya et al., 1995). Since DMS+Cl is neither a big sink of DMS nor a big source of DMSO in our study, the yield uncertainties have little influence on the modeled sulfur budgets. However, modeled estimates of DMS+Cl could be too low due to a potential low bias in modeled Cl abundance.





In this study, DMS+O$_{3(aq)}$ is the only multiphase DMS oxidation pathway, which accounts for only 1% of DMS oxidation globally, reaching up to 5% over high-latitude oceans (e.g. Southern Ocean) (Fig. 3). In comparison, in a general circulation model Boucher et a. (2003) calculated that DMS+O$_{3(aq)}$ accounts for about 6% of DMS oxidation globally and 15-30% over oceans north of 60°N and in the 50-75°S latitude band. The difference between the results from Boucher et al. (2003) and

this study is that Boucher et al. (2003) did not include DMS+BrO in their model simulation, which could lead to an overestimate of the contribution of DMS+O$_{3(aq)}$ as both reactions are most important over the high-latitude oceans during winter. Using a 1-D model, von Glasow and Crutzen (2004) calculated that DMS+O$_{3(aq)}$ accounts for 4-18% of DMS oxidation in the cloudy MBL, which is similar to 5-10% over the Southern Ocean MBL in our model results. The fraction of DMS oxidized by O$_3$ in the gas phase ($f_{[l]DMS+O3(g)}$=0.5%) is only half of $f_{[l]DMS+O3(aq)}$, consistent with Boucher et al. (2003).

Thus, both the gas-phase and multiphase oxidation of DMS by O$_3$ represent minor DMS sinks in the global troposphere.

### 3.2 DMSO budget

The modeled global tropospheric DMSO burden is 8 Gg S, which is 3-4 times larger than in Pham et al. (1995) and Cosme et al. (2002) which did not include production of DMSO from DMS+BrO. Modeled surface DMSO mixing ratio is highest over Southern Ocean ($\approx$30 ppt) (Fig. 2b) where DMS mixing ratio is high and BrO is abundant. The high DMSO mixing ratio

over Antarctica in our model is due to weak DMSO oxidation by OH in both the gas and aqueous phase. DMSO mainly resides in the lower troposphere, with 66% of the tropospheric burden below 2 km.

Globally, we simulate DMS+BrO is the biggest source of DMSO (45%), followed by the addition channel of DMS+OH (42%), DMS+Cl (9%) and DMS+O$_{3(aq)}$ (4%). The fraction of DMSO produced from DMS+BrO is highest over high-latitude

ocean where OH abundance is low and subtropical oceans where BrO abundance is high, while DMS+Cl and DMS+O$_{3(aq)}$ can account for up to 20% of DMSO production in coastal regions and mid-latitude MBL, respectively (Fig. 4).

DMSO is removed from the atmosphere via gas-phase oxidation by OH (37%), multiphase oxidation by OH in cloud droplets (31%) and aerosols (3%), and deposition (29%). The lifetime of DMSO is about 12 hours. Multiphase oxidation is

especially important over regions where clouds are frequent and OH concentrations are high, e.g. low- to mid-latitude oceans (Fig. 5). Cosme et al. (2002) calculated 85% of DMSO is lost via gas-phase oxidation by OH and the rest 15% via deposition in a global 3-D model, but they did not include heterogeneous loss of DMSO. It has been suggested that heterogeneous loss is the predominant loss process of DMSO in the cloudy MBL in box or 1-D models (Zhu et al., 2006; Hoffmann et al., 2016).

### 3.3 MSIA budget

MSIA is an important intermediate during the oxidation of DMSO to produce MSA, and has a simulated tropospheric burden of 4 Gg S. The surface MSIA mixing ratio is higher over Antarctica than Southern Ocean (Fig. 2c) due to larger removal of





MSIA by $O_{3(aq)}$ and $OH_{(aq)}$ in clouds over Southern Ocean. 41% of MSIA resides below 2 km altitude.

In $R_{all}$, MISA is produced from both gas-phase (53%) and multiphase (47%) oxidation of DMSO by OH in cloud droplets and aerosols (Fig. 1). Multiphase production of MSIA is more important over low- to mid-latitude oceans where OH abundance is high and clouds are frequent (Fig. 6).

MSIA is mainly removed in the troposphere via both gas-phase and multiphase oxidation by OH, with a lifetime of 9 hours. The deposition of MSIA accounts for 8% of MSIA removal in the troposphere. Globally, multiphase oxidation in cloud droplets and aerosols by $O_{3(aq)}$ (45%) and $OH_{(aq)}$ (16%) is the biggest sink of MSIA, followed by gas-phase oxidation by OH (30%). Multiphase oxidation by $OH_{(aq)}$ is more important over low-latitude oceans where OH abundance is high, but is generally within 30% (Fig. 7). Multiphase oxidation by $O_{3(aq)}$ is more important over high-latitude ocean where OH abundance is low (Fig. 7). Over continents and polar regions, MSIA is mostly oxidized by OH in the gas phase.

In comparison, Hoffmann et al. (2016) also found that multiphase oxidation is the main sink of MSIA in the MBL in their box model, with $O_{3(aq)}$, $OH_{(aq)}$ and $Cl_2^-$ accounting for 42%, 19% and 10% of MSIA removal, respectively. The rest of MSIA (29%) was removed by $CH3SO2(O2\cdot)$ that was produced as an intermediate during electron transfer reaction of MSIA with $OH_{(aq)}$ and $Cl_2^-$ in cloud droplets and aerosols. Hoffman et al. (2016) is the only modeling study that considered multiphase reaction of MSIA with both $O_{3(aq)}$ and $CH3SO2(O2\cdot)$. Zhu et al. (2006) found $Cl_2^-$ to be more important than $OH_{(aq)}$ for MSIA oxidation when assuming $Cl_2^-$ concentration 6 times higher than that used in Hoffmann et al. (2016). Due to our limited knowledge about $CH3SO2(O2\cdot)$ and $Cl_2^-$ production and concentrations in cloud droplets and aerosols, we do not include multiphase reaction of MSIA with $CH3SO2(O2\cdot)$ and $Cl_2^-$ in this study.

Gas-phase oxidation by OH (30%) has important implications for the MSA budget as MSIA+$OH_{(g)}$ has a low yield for MSA formation ($SO_2$ yield of 0.9) (Kukui et al., 2003). Gas-phase oxidation by $O_3$ is negligible globally (1%). In contrast, Lucas and Prinn (2002) suggest MSIA+$O_{3(g)}$ could compete with MSIA+$OH_{(g)}$ for MSIA removal, but the rate coefficient of MSIA+$OH_{(g)}$ is very small in their 1-D model (about two orders of magnitude smaller than ours).

### 3.4 MSA budget

In $R_{all}$, the global MSA burden is 10 Gg S, which is lower than the range of 13-40 Gg S reported in previous modeling studies (Pham et al, 1995; Chin et al., 1996; 2000; Cosme et al., 2002; Hezel et al., 2011). The largest MSA burden is from Hezel et al. (2011), in which DMSO was not included, while the smallest MSA burden is from Cosme et al. (2002), in which DMSO was included. Neglecting the DMSO intermediate in the model could result in an overestimate of MSA production as DMSO is removed via dry and wet deposition leading to no MSA production. Note that none of these previous studies consider DMS+BrO and MSA+$OH_{(aq)}$ in their models. The global MSA burden drops to 8 Gg S in our sensitivity simulation





without DMS+BrO and increases to 15 Gg S in our sensitivity simulation without MSA+OH$_{(aq)}$, which demonstrates the important role of DMS+BrO for MSA formation and MSA+OH$_{(aq)}$ for MSA removal. Surface MSA mixing ratio is highest over the Southern Ocean, but the peak shifts north compared to DMS, DMSO and MSIA (Fig. 2d). This is due to larger production of MSA by O$_{3(aq)}$ and OH$_{(aq)}$ in clouds (due to higher O$_{3(aq)}$ and OH$_{(aq)}$ concentrations at lower latitudes) over

northern part of Southern Ocean compared to the southern part of Southern Ocean. 60% of MSA resides below 2 km altitude, suggesting that MSA is mainly produced in the MBL.

As shown in Fig. 1, MSA is mainly produced from multiphase oxidation of MSIA by O$_3$ (69%) and OH (25%). MSIA+OH$_{(aq)}$ dominates over low-latitude oceans while MSIA+O$_{3(aq)}$ dominates over high-latitude oceans (Fig. 8). MSA

formation occurs mainly in clouds (58%), where liquid water content is high. Our result is consistent with the general concept that gas-phase MSA formation is small compared to multiphase formation (Barnes et al., 2006; von Glasow and Crutzen, 2004; Zhu et al., 2006; Hoffmann et al., 2016). MSA+OH$_{(aq)}$ is an important sink of MSA in our model, accounting for 38% of MSA removal. As a result, the lifetime of MSA is only 1.4 days globally, which is relatively short compared to 5-7 days in previous studies (Pham et al, 1995; Chin et al., 1996; 2000; Cosme et al., 2002; Hezel et al., 2011) without

MSA+OH$_{(aq)}$. The MSA lifetime is lowest (about 20 hours) over tropical oceans where clouds are frequent and OH abundance is high. It increases to 2-6 days over Southern Ocean and subtropical ocean. To the best of our knowledge, this is the first study to report global MSA lifetime from a global 3-D model that considers MSA+OH$_{(aq)}$. In the sensitivity run without MSA+OH$_{(aq)}$ ($R_{\text{noMSA+OH(aq)}}$), the lifetime of MSA increases to 2 days. As shown in Sect. 3.6.2, MSA+OH$_{(aq)}$ has a significant impact on modeled MSA/nssSO$_4^{2-}$ ratio.

**3.5 Uncertainties in rate constants**

The uncertainties in the rate constants for the reactions added in the model are shown in Table 4. The uncertainty factor ($f_{298}$) used for gas-phase reaction rate constant at 298 K indicates that the reaction rate constant could be greater than or less than the recommended value by a factor of $f_{298}$. For all gas-phase reactions added in this study, $f_{298}$ varies from 1.2 to 1.5. $f_{298}$ is 1.3 for the DMS+BrO reaction, which adds to the uncertainty in oxidation of DMS by BrO. The global annual mean

tropospheric BrO burden varies from 3.6 to 5.7 Gg Br in three recent global modeling studies (Parrella et al., 2012; Schmidt et al., 2016; Chen et al., 2017), but all three of these modeling studies underestimate satellite observations of tropospheric BrO column from Theys et al. (2010) (e.g. by 44% over Southern Ocean in Chen et al. (2017)). Thus, further investigations are needed in both laboratory determination of the reaction rate constant for DMS+BrO and field observation of BrO abundance in the troposphere. In addition, we need to better constrain the rate constants for the other two gas-phase reactions

DMS+OH (addition pathway) and DMSO+OH ($f_{298}$=1.2). Very few studies have determined the rate constants for the multiphase reactions added in the model (Table 4). The biggest uncertainty resides in MSA+OH$_{(aq)}$ and MSIA+O$_{3(aq)}$ reactions. The rate constant for the MSA+OH$_{(aq)}$ reactions differs by a factor of 4.7 in Milne et al. (1989) and Zhu et al. (2003), which results in about 30% difference in global annual mean tropospheric MSA burden. Only one box modeling



study (Hoffmann et al., 2016) considered MSIA+$O_{3(aq)}$ reaction, using the rate constant measured in Herrmann and Zellner (1997). As MSIA+$O_{3(aq)}$ and MSA+$OH_{(aq)}$ account for 69% of MSA production and 38% of MSA removal, respectively, more laboratory studies are needed to constrain the rate constants for these two reactions.

### 3.6 Model-observation comparison

### 3.6.1 Surface DMS mixing ratio

Observations of monthly mean DMS mixing ratio from 4 stations around the globe are used to assess modeled DMS: Crete Island (CI; 35°24'N, 25°60'E) (Kouvarakis and Mihalopoulos, 2002), Amsterdam Island (AI; 37°50'S, 77°30'E) (Castebrunet et al., 2009), Cape Grim (CG; 40°41'S, 144°41'E) (Ayer et al., 1995), and Dumont D'Urville (DU; 66°40'S, 140°1'E) (Castebrunet et al., 2009). The DMS data covers 1997-1999 period for CI, 1987-2006 period for AI, 1989-1992

period for CG, and 1998-2006 period for DU.

Figure 9 shows the comparison between modeled and observed monthly-mean DMS mixing ratio at CI, AI, CG and DU stations. Comparing $R_{all}$ with $R_{std}$, we can see that in general the modeled DMS mixing ratios match better with observations for the three stations in the Southern Hemisphere with the updated DMS chemistry, especially during Southern Hemisphere

winter. Between June and August, the modeled DMS mixing ratios calculated from $R_{std}$ overestimate observations by a factor of 6, 4 and 27 for AI, CG and DU, respectively. In comparison, during the same period, the modeled DMS mixing ratios calculated from $R_{all}$ overestimate observations by a factor of 3 for AI, 50% for CG and a factor of 4 for DU, respectively. The smaller discrepancy between modeled and observed DMS mixing ratio in $R_{all}$ is largely due to DMS+BrO, as indicated by comparing $R_{all}$ with a model run that includes all reactions except DMS+BrO ($R_{noDMS+BrO}$). It should be noted that BrO is

underestimated in our model compared to satellite observations (underestimated by 44% in terms of annual mean tropospheric BrO column between 30°S and 60°S) (Chen et al., 2017), which at least partly explains the overestimate of DMS mixing ratios from $R_{all}$ compared to observations.

In addition to DMS chemistry shown above, surface seawater DMS concentrations also affect modeled DMS mixing ratio.

The surface seawater DMS concentration was obtained from Kettle et al. (1999) in $R_{Kettle}$, instead of from Lana et al. (2011) in $R_{all}$. The global DMS emission flux from $R_{Kettle}$ is 15% lower than that from $R_{all}$. Overall, at CI, CG and DU, the modeled DMS mixing ratios from $R_{Kettle}$ are similar to those from $R_{all}$ during most of the year. Much lower DMS mixing ratios were calculated from $R_{Kettle}$ at CI in June, at CG in January and at DU in December and January. At AI, however, the modeled DMS mixing ratios from $R_{Kettle}$ are lower than those from $R_{all}$ in general, which agree better with observations except in

December and January. In this study, we focus on the chemistry aspects of the sulfur cycle and thus will not present further discussion on the impact of DMS sea water climatology on atmospheric DMS abundance.



### 3.6.2 Surface MSA/nssSO$_4^{2-}$ ratio

Figure 10 shows the comparison between modeled and observed annual-mean MSA/nssSO$_4^{2-}$ ratio at 23 stations around the globe: Dye (DY; 66°N, 53°E), Heimaey (HE; 63°N, 20°W), United Kingdom (UK; 58°N, 6°W), Mace Head (MH; 53°N, 10°W), Crete Island (CI; 35°N, 25°E), Bermuda (BE; 32°N, 65°W), Tenerife (TE; 28°N, 17°W), Midway Island (MI; 28°N,

177°W), Miami (MI; 26°N, 80°W), Barbados (BA; 13°N, 60°W), Fanning Island (FI; 4°N, 159°W), American Samoa (AS; 14°S, 170°W), New Caledonia (NC; 21°S, 166°E), Norfolk Island (NI; 29°S, 168°E), Amsterdam Island (AI; 38°S, 77°E), Cape Grim (CG; 40°S, 144°E), Palmer (PA; 65°S, 64°W), Dumont D'Urville (DU; 66°S, 140°E), Mawson (MA; 67°S, 63°E), Neumayer (NE; 70°S, 8°W), Halley Bay (HB; 75°S, 26°W), Kohnen (KO; 75°S, 0°E) and Dome C (DC; 75°S, 123°E). Data for all stations was obtained from Gondwe et al. (2004), except for CI from Kouvarakis and Mihalopoulos

(2002) and AI, PA, KO and DC from Casterbrunet et al. (2009). The global distribution of annual-mean MSA/nssSO$_4^{2-}$ obtained from $R_\text{all}$, overplotted with observations for these 23 stations are shown in Fig. 11. In addition to the 4 model runs described in Sect. 3.6.1 ($R_\text{alll}$, $R_\text{std}$, $R_\text{Kettle}$ and $R_\text{noDMS+BrO}$), 4 additional model runs were performed by removing ($R_\text{noMSA+OH(aq)}$) or decreasing ($R_\text{lessMSA+OH(aq)}$) aqueous-phase oxidation of MSA by OH, removing all multiphase chemistry involving DMS, DMSO, MSIA and MSA oxidation ($R_\text{noMUL}$), and decreasing OH$_\text{(aq)}$ concentrations in cloud droplets and

aerosols by two orders of magnitude ($R_\text{lowOH(aq)}$).

Figures 10 and 11 show that modeled MSA/nssSO$_4^{2-}$ ratios calculated from $R_\text{all}$ are in good agreement with observations, with a normalized mean bias $N_\text{MB}$ ($= \frac{\sum_{i=1}^{23}(M_i - O_i)}{\sum_{i=1}^{23} O_i} \times 100\%$, where $M_i$ and $O_i$ are modeled value and observed value, respectively) of 19%. The model reproduces the spatial variability of MSA/nssSO$_4^{2-}$ observations very well for Northern

Hemisphere stations where anthropogenic sources of SO$_2$ dominate nssSO$_4^{2-}$ abundance. Over low-latitude oceans (13°N-37°S), modeled ratios overestimate observations, but still capture the latitudinal trend of increasing ratios towards the south where anthropogenic sources of nssSO$_4^{2-}$ are less important. The large modeled MSA/nssSO$_4^{2-}$ over low-latitude oceans is due to lower anthropogenic sources of nssSO$_4^{2-}$ and to large multiphase MSA production as a result of high cloud liquid water content and oxidant abundance (OH and O$_3$). Over Antarctica (Stations PA, DU, MA, NE, HB, KO and DC), though

$R_\text{all}$ is not able to reproduce the spatial variability of observations, it shows good agreement with observations on average (MSA/nssSO$_4^{2-}$ is 0.24 for observations and 0.22 for $R_\text{all}$). In $R_\text{noDMS+BrO}$, modeled MSA/nssSO$_4^{2-}$ ratios decrease compared to $R_\text{all}$, which is most evident over stations where DMS+BrO is a large source of DMSO and MSA (e.g. Southern Hemisphere ocean) (Fig. 4). If multiphase chemistry is switched off ($R_\text{noMUL}$), modeled MSA/nssSO$_4^{2-}$ ratios underestimate the observations by 49% on average for all 23 stations. Thus, multiphase sulfur chemistry is important for the model simulation

of MSA/nssSO$_4^{2-}$ observations. However, the OH$_\text{(aq)}$ concentrations in cloud droplets and aerosols, which range from 10$^{-14}$ M to 10$^{-12}$ M in modeling studies (Jacob, 1986; Matthijsen et al., 1995; Jacob et al., 1989; Herrmann et al., 2000) and 10$^{-16}$ M to 10$^{-14}$ M in observations (Arakaki et al., 2013; Kaur and Anastasio, 2017), is a large uncertainty in modeling multiphase sulfur chemistry. The model run reducing OH$_\text{(aq)}$ concentrations by two orders of magnitude ($R_\text{lowOH(aq)}$) results in an increase



in MSA/nssSO$_4^{2-}$, which overestimates MSA/nssSO$_4^{2-}$ observations by 92% on average. Model runs without MSA+OH$_{(aq)}$ ($R_{noMSA+OH(aq)}$) or with smaller reaction rate coefficient of MSA+OH$_{(aq)}$ ($R_{lessMSA+OH(aq)}$) result in an increase of modeled MSA/nssSO$_4^{2-}$ compared to $R_{all}$, which largely overestimates MSA/nssSO$_4^{2-}$ observations at low- to mid-latitudes MBL between 28°N and 37°S. This reveals the importance of MSA+OH$_{(aq)}$ for the interpretation of MSA/nssSO$_4^{2-}$ observations, as

suggested by von Glasow and Crutzen (2004), Zhu et al. (2006) and Mungall et al. (2018).

Modeled MSA/nssSO$_4^{2-}$ from $R_{std}$ without multiphase chemistry and DMS+BrO can generally reproduce the meridional trend of observations, with $N_{MB}$=51%. However, $R_{std}$ overestimates DMS observations (Fig. 9), suggesting that $R_{std}$ produces comparable MSA/nssSO$_4^{2-}$ values for the wrong reasons.

**4 Implications**

Once emitted into the atmosphere through air-sea exchange, biogenic DMS undergoes complicated chemical processes to form SO$_2$ and MSA in the troposphere. SO$_2$ can then be oxidized to form sulfate aerosol. Sulfate and MSA produced in the gas phase can nucleate new particles under favorable conditions (Kulmala et al., 2000; Chen et al., 2015), while MSA and sulfate produced in the aqueous phase leads to the growth of existing particles (Kreidenweis and Seinfeld, 1988; Kaufman

and Tanre, 1994). Global models such as General Circulation Models (GCMs) and Chemical Transport Models (CTMs) generally consider very simplified gas-phase DMS chemistry, which could result in large biases in SO$_2$ and MSA prediction. Quantifying the yields of SO$_2$ and MSA from DMS oxidation is necessary to evaluate the climate impacts of DMS from the ocean ecosystem. Compared to the standard GEOS-Chem model run, the updated sulfur scheme in this study decreases the conversion yield of DMS to SO$_2$ ($Y_{DMS \rightarrow SO2}$) from 91% to 78% and increases the conversion yield of DMS to MSA

($Y_{DMS \rightarrow MSA}$) from 9% to 13%. In order to gain insight into the impacts of our updated sulfur scheme on global SO$_2$, MSA and sulfate burden, we conducted two sensitivity studies by allowing DMS as the only sulfur source for both the standard model run $R_{std}$ ($R_{std\_onlyDMS}$) and full model run $R_{all}$ ($R_{all\_onlyDMS}$). Compared to $R_{std\_onlyDMS}$, the global DMS, SO$_2$, MSA and sulfate burden in $R_{all\_onlyDMS}$ decrease by 39%, 15%, 55% and 2%, respectively. The decrease in DMS is mainly due to DMS oxidation by BrO with the updated sulfur scheme. The decrease in SO$_2$ is due to a lower yield of SO$_2$ from DMS ($Y_{DMS \rightarrow SO2}$),

but is partly compensated by the increase in the DMS oxidation rate. MSA decreases despite an increase in the yield of MSA from DMS ($Y_{DMS \rightarrow MSA}$) due to the inclusion of MSA+OH$_{(aq)}$. The decrease in sulfate is relatively small, mostly due to the inclusion of MSA+ OH$_{(aq)}$ as a sulfate source, which accounts for 9% of global sulfate production. In sum, climate models with a simplified DMS oxidation scheme (gas-phase oxidation by OH and NO$_3$ only) may overestimate SO$_2$, MSA and sulfate abundances in the pre-industrial environment, potentially leading to underestimates in sulfur aerosol radiative forcing

calculations in climate models.





MSA in Antarctic ice cores has been related to spring sea ice extent (Curran et al., 2003; Abram et al., 2010) as DMS is emitted in regions of sea ice melt. Our results show that, in addition to DMS emission, tropospheric sulfur chemistry is critical for MSA abundance in the troposphere. Compared to the full model run $R_{all}$, sensitivity studies without DMS+BrO reaction ($R_{noDMS+BrO}$) and without multiphase oxidation of DMS, DMSO, MSIA and MSA ($R_{noMUL}$) reduce global MSA

burden by 20% and 51%, respectively. This indicates that reactive halogen and multiphase chemistry are important for the MSA budget in the troposphere, which should be considered when interpreting MSA abundance in ice cores, especially over time periods where the abundance of atmospheric oxidants may have changed.

## 5 Conclusions

In this study, we investigate the impacts of reactive halogen and multiphase chemistry on tropospheric DMS chemistry by

adding 2 new chemical tracers (DMSO and MSIA) and 12 new reactions for both the gas-phase and multiphase oxidation of DMS, DMSO, MSIA and MSA into a global chemical transport model, GEOS-Chem. With the updated DMS chemistry, the DMS burden decreases by 40% globally, mostly due to oxidation of DMS by BrO. BrO oxidation accounts for 12% of DMS oxidation globally, which could be underestimated due to underestimates in BrO abundance in the model, but is within the range of 8-16% reported in previous studies. Cl is not important for DMS oxidation due to small Cl abundance, but this

reaction should be revisited if modeled Cl budgets are substantially revised in the future. Both gas-phase and multiphase oxidation of DMS by $O_3$ are not important for the global DMS budget and can be neglected in global models.

Dry and wet deposition accounts for 29% of DMSO removal and 8% of MSIA removal globally. The significant role of deposition as a sink for DMSO suggests that DMSO should be included in sulfur chemistry mechanisms, as exclusion of

DMSO as an intermediate may result in an overestimate of MSA production from the oxidation of DMS. MSIA is an important intermediate between DMSO and MSA. MSA is mostly (94% globally) produced through aqueous phase oxidation of MSIA by $O_{3(aq)}$ and $OH_{(aq)}$ in cloud droplets and aerosols. Dry and wet deposition accounts for 62% of MSA removal globally, multiphase oxidation by OH in cloud droplets and aerosols accounts for the rest. We note that the relative importance of deposition versus oxidation as a sink for MSA is sensitive to the $OH_{(aq)}$ concentration in cloud droplets and

aerosols, which is highly uncertain.

Modeled DMS mixing ratios agree better (mean square error between model and observation is 44% smaller) with observations with the inclusion of DMS+BrO, and modeled MSA/nssSO$_4^{2-}$ ratios agree better (mean square error between model and observation is 21% smaller) with observations with the inclusion of DMS+BrO and multiphase chemistry

associated with DMS, DMSO, MSIA and MSA oxidation. The uncertainties of reactive halogen abundances such as BrO and Cl and the aqueous phase oxidant concentrations such as $OH_{(aq)}$ have limited our ability to model DMS oxidation and MSA




formation in the troposphere. Future studies should prioritize the measurements of reactive halogen abundances and $OH_{(aq)}$ concentrations in cloud droplets, especially in the marine boundary layer.

**Acknowledgements**

We acknowledge Dr. Johan A. Schmidt for the discussions on tropospheric sulfur and halogen cycles, and providing support for the GEOS-Chem halogen chemistry scheme. We acknowledge Dr. Tom Breider for offering help on DMSO simulation scheme. We acknowledges Dr. Paul Hezel for the help on searching for sulfur species observations, and Dr. Giorgos Kouvarakis for providing DMS data on Crete Island. We acknowledge NSF AGS award 1343077 to B.A. for support of this research.

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





**Figures**

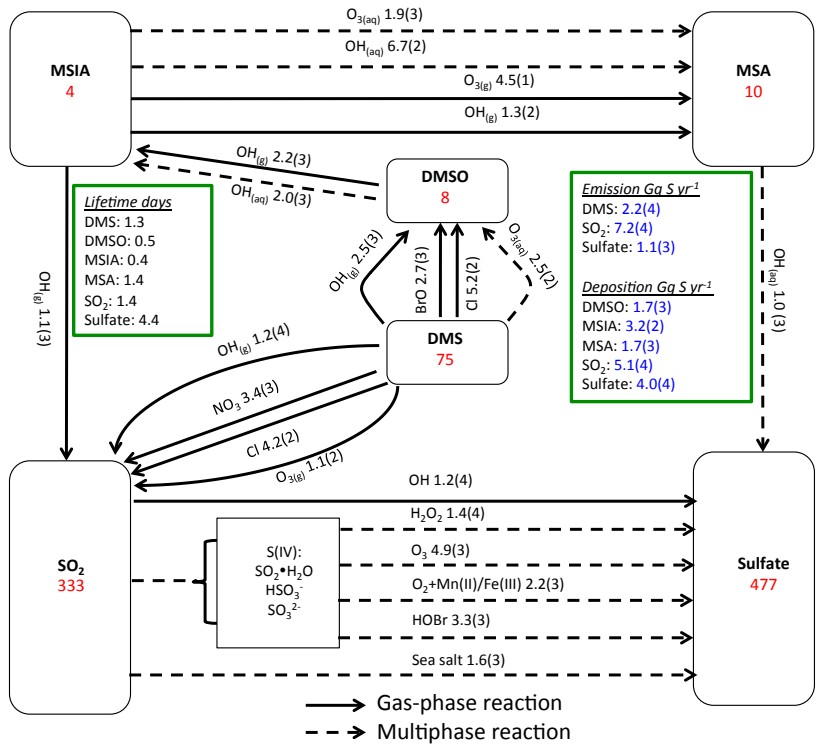

**Figure 1: Global sulfur budgets for $R_{all}$. Inventories (inside the boxes) are in units of Gg S. Solid arrows represent gas-phase reactions while dashed arrows represent aqueous-phase reactions. Production and loss rates above arrows are in unit of Gg S yr$^{-1}$.**
5 **Read 1.9(3) as 1.9×10$^3$ Gg S yr$^{-1}$.**



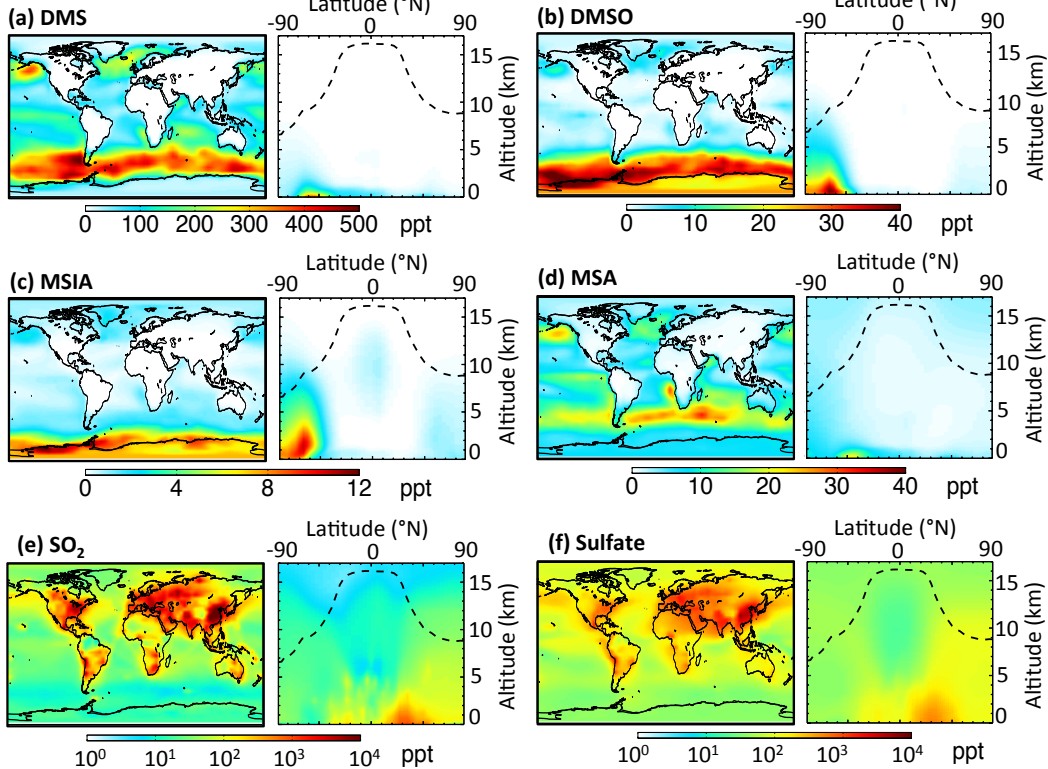

**Figure 2: Horizontal distribution of annual-mean surface mixing ratios (ppt) and vertical distribution of mixing ratios for (a) DMS, (b) DMSO, (c) MSIA, (d) MSA, (e) SO₂ and (f) sulfate. The dashed line indicates tropopause height.**





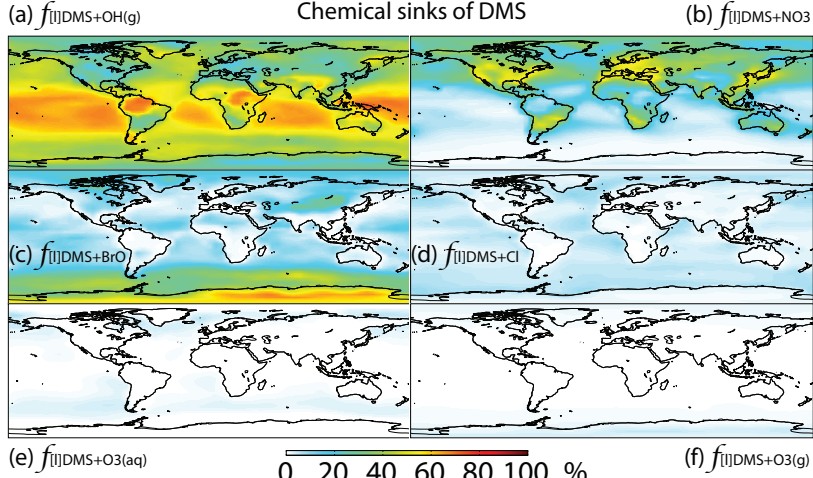

**Figure 3: Global tropospheric distribution of annual-mean percentage of DMS oxidized in the troposphere via (a) DMS+OH$_{(g)}$ ($f_{[l]DMS+OH(g)}$), (b) DMS+NO$_3$ ($f_{[l]DMS+NO3}$), (c) DMS+BrO ($f_{[l]DMS+BrO}$), (d) DMS+Cl ($f_{[l]DMS+Cl}$), (e) DMS+O$_{3(aq)}$ ($f_{[l]DMS+O3(aq)}$) and (f) DMS+O$_{3(g)}$ ($f_{[l]DMS+O3(g)}$).**

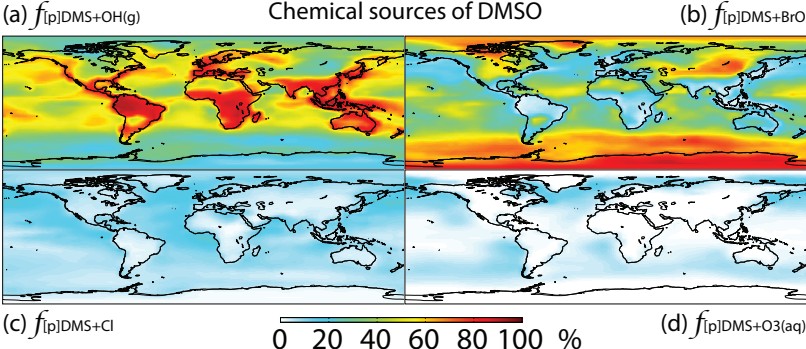

**Figure 4: Global tropospheric distribution of annual-mean percentage of DMSO produced via (a) DMS+OH$_{(g)}$ ($f_{[p]DMS+OH(g)}$), (b) DMS+BrO ($f_{[p]DMS+BrO}$), (c) DMS+Cl ($f_{[p]DMS+Cl}$) and (d) DMS+O$_{3(aq)}$ ($f_{[p]DMS+O3(aq)}$).**



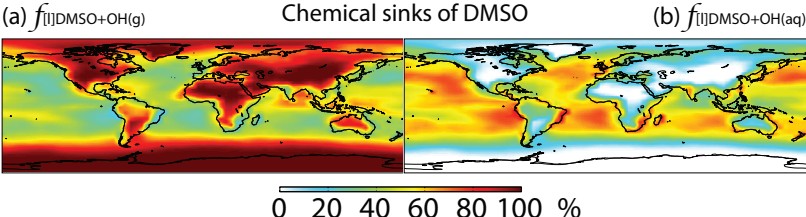

**Figure 5: Global tropospheric distribution of annual-mean percentage of DMSO oxidized via (a) DMSO+OH$_{(g)}$ ($f_{[l]DMSO+OH(g)}$) and (b) DMSO+OH$_{(aq)}$ ($f_{[l]DMSO+OH(aq)}$).**

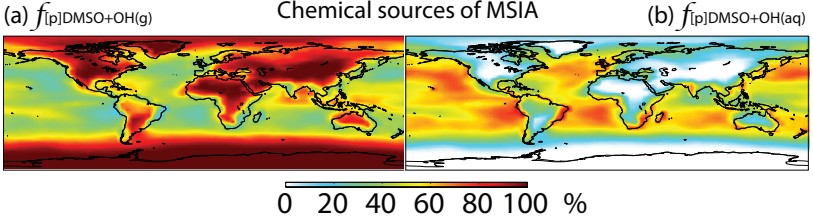

5    **Figure 6: Global tropospheric distribution of annual-mean percentage of MSIA produced in the troposphere via (a) DMSO+OH$_{(g)}$ ($f_{[p]DMSO+OH(g)}$) and (b) DMSO+OH$_{(aq)}$ ($f_{[p]DMSO+OH(aq)}$).**

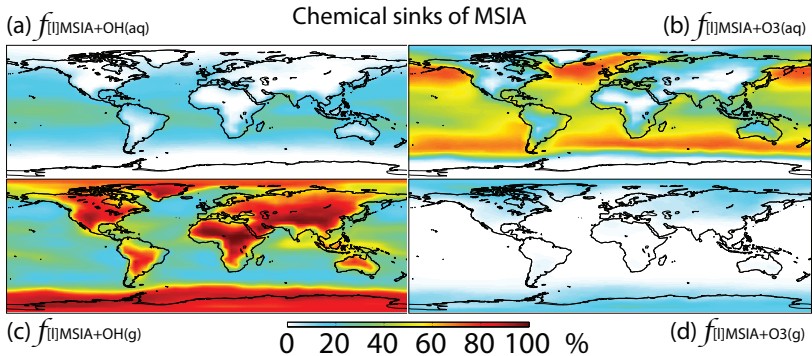

**Figure 7: Global tropospheric distribution of annual-mean percentage of MSIA oxidized in the troposphere via (a) MSIA+OH$_{(aq)}$**
10    **($f_{[l]MSIA+OH(aq)}$), (b) MSIA+O$_{3(aq)}$ ($f_{[l]MSIA+O3(aq)}$), (c) MSIA+OH$_{(g)}$ ($f_{[l]MSIA+OH(g)}$) and (d) MSIA+O$_{3(g)}$ ($f_{[l]MSIA+O3(g)}$).**


Chemical sources of MSA

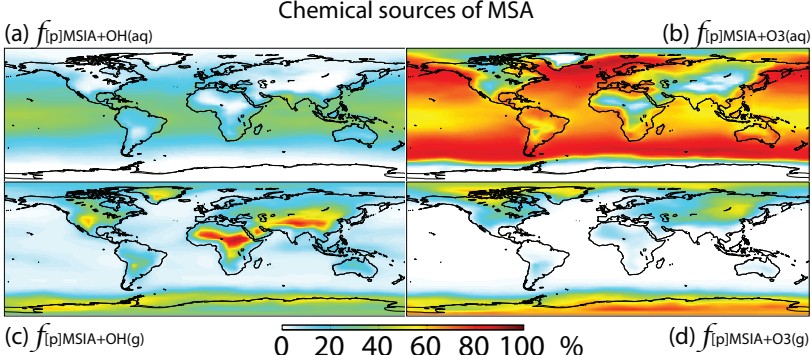

**Figure 8: Global tropospheric distribution of annual-mean percentage of MSA produced in the troposphere (a) MSIA+OH$_{(aq)}$ ($f_{[p]MSIA+OH(aq)}$), (b) MSIA+O$_{3(aq)}$ ($f_{[p]MSIA+O3(aq)}$), (c) MSIA+OH$_{(g)}$ ($f_{[p]MSIA+OH(g)}$) and (d) MSIA+O$_{3(g)}$ ($f_{[p]MSIA+O3(g)}$).**

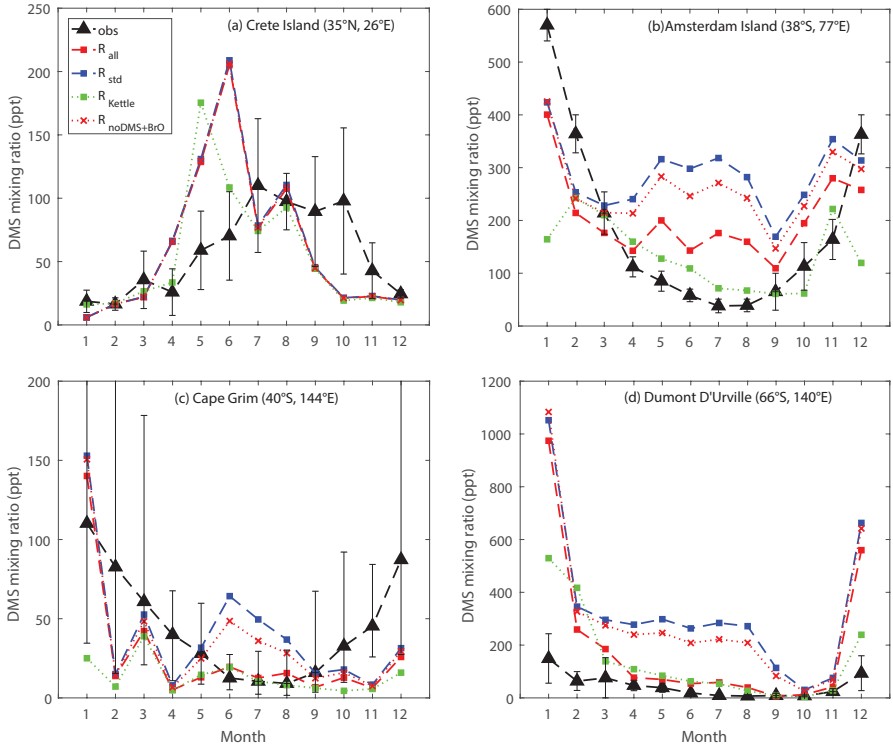

5    **Figure 9: Comparison between modeled and observed monthly mean surface DMS mixing ratios at (a) Crete Island (CI), (b) Amsterdam Island (AI), (c) Cape Grim (CG), and (d) Dumont D'Urville (DU) stations.**





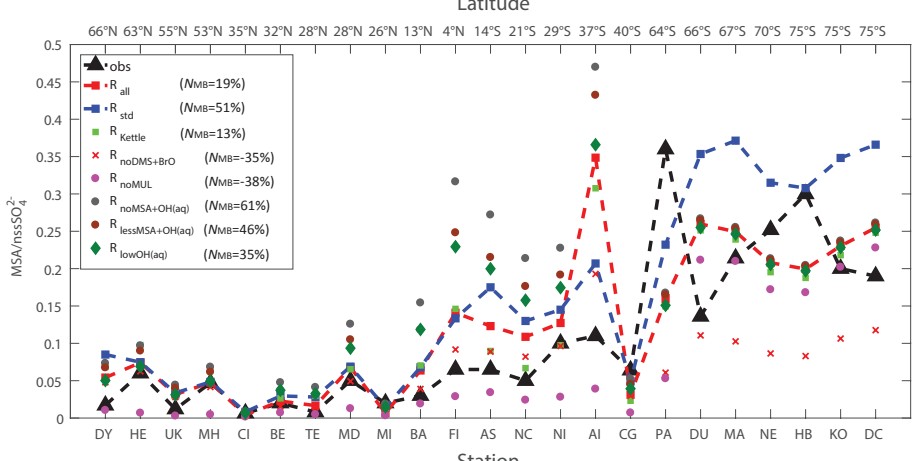

**Figure 10: Comparison between modeled (eight model runs described in Table 3) and observed (obs, black triangle) annual mean surface MSA/nssSO$_4^{2-}$ ratios at 23 stations around the globe. The normalized mean bias $N_{MB} = \frac{\sum_{i=1}^{23}(M_i - O_i)}{\sum_{i=1}^{23} O_i} \times 100\%$, where $M_i$ and $O_i$ are modeled value and observed value, respectively, is shown in inset.**

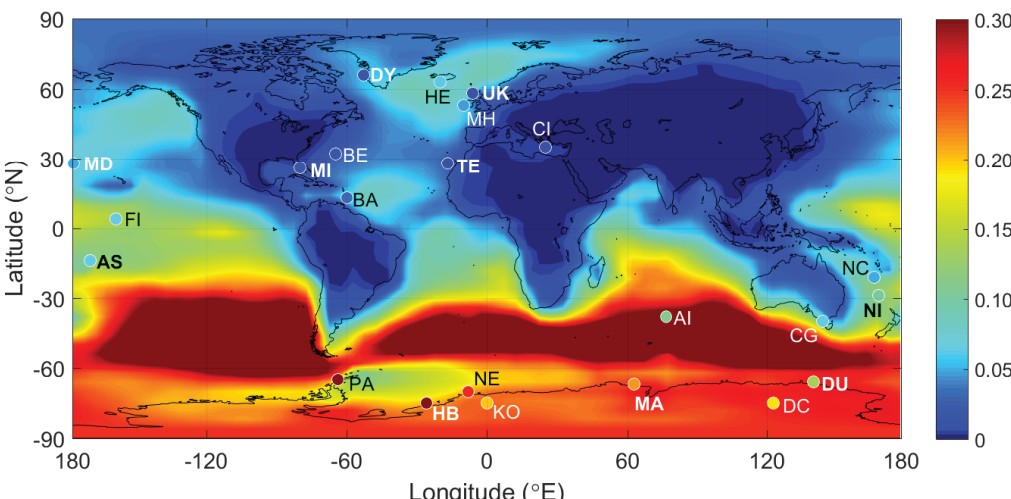

**Figure 11: Global distribution of annual mean surface MSA/nssSO$_4^{2-}$ molar ratios from the full model run ($R_{all}$), overplotted with observed annual mean surface MSA/nssSO$_4^{2-}$ ratios from 23 stations around the globe.**




**Tables**

**Table 1.** Overview of sulfur chemistry in the full model run ($R_{all}$) with DMSO and MSIA intermediates and all 12 new reactions.

| Gas-phase reactions | $k_{298}$ [cm$^3$ s$^{-1}$] | $-E_a/R$ [K] | Reference |
|---|---|---|---|
| DMS+OH $\xrightarrow{abstraction}$ SO$_2$+CH$_3$O$_2$ + CH$_2$O | $4.69\times10^{-12}$ | -280 | Burkholder et al. (2015) |
| DMS+OH $\xrightarrow{addition}$ 0.6SO$_2$+0.4DMSO+CH$_3$O$_2$[new] | see note[a] | | Burkholder et al. (2015); Pham et al. (1995); Spracklen et al. (2005) |
| DMS+NO$_3$ → SO$_2$+HNO$_3$+CH$_3$O$_2$+CH$_2$O | $1.13\times10^{-12}$ | 530 | Burkholder et al. (2015) |
| DMS+BrO → DMSO+Br[new] | $3.39\times10^{-13}$ | 950 | Burkholder et al. (2015) |
| DMS+O$_3$ → SO$_2$[new] | $1.00\times10^{-19}$ | 0 | Burkholder et al. (2015); Du et al. (2007) |
| DMS+Cl → 0.5SO$_2$+0.5DMSO+0.5HCl+0.5ClO[new] | $3.40\times10^{-10}$ | 0 | Burkholder et al. (2015); Barns et al. (2006); IUPAC[e] |
| DMSO+OH→ 0.95MSIA[new] | $8.94\times10^{-11}$ | 800 | Burkholder et al. (2015); von Glasow and Crutzen (2004) |
| MSIA+OH→ 0.9SO$_2$+0.1MSA[new] | $9.0\times10^{-11}$ | 0 | Burkholder et al. (2015); Kukui et al. (2003); Hoffmann et al. (2016); Zhu et al. (2006) |
| MSIA+O$_3$→ MSA[new] | $2.0\times10^{-18}$ | 0 | Lucas and Prinn (2002); von Glasow and Crutzen (2004) |
| SO$_2$+OH $\xrightarrow{O2,H2O}$ H$_2$SO$_4$+HO$_2$ | see note[b] | | Burkholder et al. (2015) |

| multiphase reactions | $k_{298}$ [M$^{-1}$ s$^{-1}$] | $-E_a/R$ [K] | Reference |
|---|---|---|---|
| DMS$_{(g)}$+O$_{3(aq)}$ → DMSO$_{(aq)}$+O$_{2(aq)}$ [new] | $8.61\times10^8$ | -2600 | Gershenzon et al. (2001) |
| DMSO$_{(g)}$+OH$_{(aq)}$ → MSIA$_{(aq)}$ [new] | $6.63\times10^9$ | -1270 | Zhu et al. (2003) |
| MSIA$_{(g)}$+OH$_{(aq)}$→MSA$_{(aq)}$ [new] | $1.20\times10^{10}$ | 0 | Bardouki et al. (2002) |
| MSIA$_{(g)}$+O$_{3(aq)}$→MSA$_{(aq)}$ [new] | $3.50\times10^7$ | 0 | Hoffmann et al. (2016) |
| MSA$_{(g)}$+OH$_{(aq)}$ → SO$_4^{2-}$ [new] | $6.1\times10^7$ | -2630 | Milne et al. (1989); Zhu et al. (2003); Barnes et al. (2006) |
| HSO$_3^-$+H$_2$O$_{2(aq)}$+H$^+$ → SO$_4^{2-}$+2H$^+$+H$_2$O$_{(aq)}$ | $2.36\times10^{3[c]}$ | -4760 | Jacob (1986) |
| HSO$_3^-$+O$_{3(aq)}$→ SO$_4^{2-}$+H$^+$+O$_{2(aq)}$ | $3.20\times10^5$ | -4830 | Jacob (1986) |



| | | | |
|---|---|---|---|
| $SO_3^{2-} + O_{3(aq)} \rightarrow SO_4^{2-} + O_{2(aq)}$ | $1.00 \times 10^9$ | -4030 | Jacob (1986) |
| $S(IV) + O_{2(aq)} \xrightarrow{Mn(II), Fe(III)} SO_4^{2-}$ | see note[(d)] | | Martin and Good (1991) |
| $HSO_3^- + HOBr_{(aq)} \rightarrow SO_4^{2-} + 2H^+ + Br^-$ | $3.20 \times 10^9$ | 0 | Liu (2000) |
| $SO_3^{2-} + HOBr_{(aq)} \rightarrow SO_4^{2-} + H^+ + Br^-$ | $5.00 \times 10^9$ | 0 | Troy and Margerum (1991) |

[(new)] New reaction added in the model.

[(a)] $k(T, [O_2], [M]) = 8.2 \times 10^{-39}[O_2]e^{5376/T}/(1 + 1.05 \times 10^{-5}([O_2]/[M])e^{3644/T})$ cm$^3$ molecule$^{-1}$ s$^{-1}$.

[(b)] low pressure limit: $3.3 \times 10^{-31}(300/T)^{4.3}$ cm$^6$ molecule$^{-2}$ s$^{-1}$; high pressure limit: $1.6 \times 10^{-12}$ cm$^3$ molecule$^{-1}$ s$^{-1}$.

[(c)] Rate constant between $HSO_3^- + H_2O_{2(aq)}$ at pH=4.5.

5  [(d)] The metal-catalyzed sulfate production rate is calculated from the following expression:

$$-\frac{d[SO_4^{2-}]}{dt} = 750[Mn(II)][S(IV)] + 2600[Fe(III)][S(IV)] + 1.0 \times 10^{10}[Mn(II)][Fe(III)][S(IV)]$$

Detailed description about [Mn(II)] and [Fe(III)] concentrations can be found in Alexander et al. (2009).

[(e)] IUPAC: http://iupac.pole-ether.fr/htdocs/datasheets/pdf/SOx13_Cl_CH3SCH3.pdf

**Table 2.** Henry's law constant at 298 K ($H_{X(298)}$), mass accommodation coefficient ($\alpha_b$) and aqueous-phase diffusivity at 298 K ($D_{l(298K)}$) for DMS, DMSO, MSIA and MSA.

| | $H_{X(298)}$ [M atm$^{-1}$] | -ΔH/R [K] | Reference | $\alpha_b$ | Reference | $D_{l(298K)}$ [m$^2$ s$^{-1}$] | Reference |
|---|---|---|---|---|---|---|---|
| DMS | 0.56 | -4480 | Campolongo et al. (1999) | 0.001 | Zhu et al. (2006) | $1.5 \times 10^{-5}$ | Saltzman et al. (1993) |
| DMSO | $1 \times 10^7$ | -2580 | Campolongo et al. (1999) | 0.1 | Zhu et al. (2006) | $1.0 \times 10^{-5}$ | Zhu et al. (2003) |
| MSIA | $1 \times 10^8$ | -1760 | Campolongo et al. (1999) | 0.1 | Zhu et al. (2006) | $1.2 \times 10^{-5}$ | Same as MSA |
| MSA | $1 \times 10^9$ | -1760 | Campolongo et al. (1999) | 0.1 | Zhu et al. (2006) | $1.2 \times 10^{-5}$ | Schweitzer et al. (1998) |



**Table 3.** Overview of model runs.

| Model run | Specification |
|---|---|
| $R_{all}$ | Full model run including all reactions described in Table 1, including the DMSO and MSIA intermediates; sea surface water DMS concentration obtained from Lana et al. (2011) |
| $R_{std}$ | Standard run which includes gas-phase oxidation of DMS by OH and $NO_3$ only, with no DMSO or MSIA intermediates |
| $R_{Kettle}$ | $R_{all}$; sea surface water DMS concentration obtained from Kettle et al. (1999) |
| $R_{noDMS+BrO}$ | $R_{all}$; without DMS+BrO reaction |
| $R_{noMUL}$ | $R_{all}$; without multiphase oxidation of DMS, DMSO, MSIA and MSA |
| $R_{noMSA+OH(aq)}$ | $R_{all}$; without MSA+$OH_{(aq)}$ reaction |
| $R_{lessMSA+OH(aq)}$ | $R_{all}$; $k_{MSA+OH(aq)}/4.7$ (Zhu et al., 2003) |
| $R_{lowOH(aq)}$ | $R_{all}$; reduce $OH_{(aq)}$ concentrations in cloud droplets and aerosols by a factor of 100 |
| $R_{all\_onlyDMS}$ | $R_{all}$; DMS emission from the ocean is the only sulfur source |
| $R_{std\_onlyDMS}$ | $R_{std}$; DMS emission from the ocean is the only sulfur source |



**Table 4.** The uncertainties of the rate constants for the 12 reactions added in the model. The uncertainty factor $f_{298}$ means the reaction rate constant may be greater than or less than the recommended value by the factor $f_{298}$. Type "R", "L" and "M" represents values obtained from " literature reviews", "laboratory measurements" and "modeling studies", respectively.

| Gas-phase reactions | $f_{298}$ | Type | Reference |
|---|---|---|---|
| DMS+OH $\xrightarrow{addition}$ ... | 1.2 | R | Burkholder et al. (2015) |
| DMS+BrO → ... | 1.3 | R | Burkholder et al. (2015) |
| DMS+O$_3$ → ... | 1.2 | L | Du et al. (2007) |
| DMS+Cl → ... | 1.2 | R | Burkholder et al. (2015) |
| DMSO+OH→ ... | 1.2 | R | Burkholder et al. (2015) |
| MSIA+OH→ ... | 1.4 | R | Burkholder et al. (2015) |
| MSIA+O$_3$→ ... | 1.5 | M | Lucas and Prinn (2002) |
| multiphase reactions | $k_{298}$ [M$^{1-n}$ s$^{-1}$] | Type | Reference |
| DMS$_{(g)}$+O$_{3(aq)}$ → ... | $(8.6\pm8.1)\times10^8$ | L | Gershenzon et al. (2001) |
|  | $(6.1\pm2.4)\times10^8$ | L | Lee and Zhou (1994) |
| DMSO$_{(g)}$+OH$_{(aq)}$ → ... | $(6.6\pm0.7)\times10^9$ | L | Zhu et al. (2003) |
|  | $7.5\times10^9$ | M | Hoffmann et al. (2016) |
|  | $(4.5\pm0.4)\times10^9$ | L | Bardouki et al. (2002) |
|  | $(5.4\pm0.3)\times10^9$ | L | Milne et al. (1989) |
| MSIA$_{(g)}$+OH$_{(aq)}$→ ... | $(1.2\pm0.2)\times10^{10}$ | L | Bardouki et al. (2002) |
|  | $7.7\times10^9$ | M | Zhu et al. (2006) |
|  | $(6.0\pm1.0)\times10^9$ | L | Sehested and Holcman (1996) |
| MSIA$_{(g)}$+O$_{3(aq)}$→ ... | $3.5\times10^7$ | M | Hoffmann et al. (2016) |
| MSA$_{(g)}$+OH$_{(aq)}$ → ... | $(6.1\pm1.1)\times10^7$ | L | Milne et al. (1989); Zhu et al. (2003) |
|  | $(1.3\pm0.1)\times10^7$ | L | Zhu et al. (2003) |

