# Peer review of "DMS oxidation and sulfur aerosol formation in the marine troposphere: a focus on reactive halogen and multiphase chemistry"

_Atmospheric Chemistry and Physics, 2018_

## Referee Comment (RC1) · Anonymous Referee #1 · 4 Jun 2018

The present paper represents a first 3D simulation of the dimethyl sulphide (DMS) oxidation by multiphase chemistry processes in the troposphere into $SO_2$ and methane sulfonic acid (MSA). For this purpose, the authors apply the global chemical transport model GEOS-Chem and have strictly revised the implemented DMS oxidation scheme following recent insights in tropospheric multiphase DMS oxidation. Therefore, the authors include 12 new reactions describing the oxidation in the gas (7) and aqueous phase (5). For the first time, the aqueous-phase oxidation of DMS oxidation products are treated in a global simulation. Therefore, two new important oxidation intermediates are implemented, dimethyl sulfoxide (DMSO) and methane sulphinic acid (MSIA). While DMSO was already implemented in other global modelling studies, the oxidation of MSIA is dealt in a global chemical transport model for the first time to the reviewer knowledge.

The results show that multiphase chemistry processes decrease the yield of $SO_2$ and increases the yield of MSA. Multiphase chemistry processes are important for a better representation of measured DMS concentrations and MSA/nss-$SO_4^{2-}$ ratios.

The paper is well-written and organized. The implementation of multiphase chemistry, the mechanism and the technical implementation is well described. The discussion of the results is expedient and understandable.

However, the mechanism has to be strictly revised to be consistent with the current box and 1D modelling studies as well as laboratory results.

**General comments**

1) The oxidation of DMS by OH happens by two pathways: H-abstraction from the methyl group and OH-addition onto the sulphur atom. The H-abstraction pathway leads predominantly to $SO_2$. The OH-addition pathway leads predominantly to DMSO (Barnes et al. 2006). The DMSO is then further oxidised into MSIA and subsequently into MSA or $SO_2$ (see von Glasow and Crutzen, 2004; Zhu et al., 2006; Barnes et al. 2006; Hoffmann et al., 2016). A direct conversion of DMS into $SO_2$ is unlikely and only convenient if the OH-addition pathway is parameterised. As the present study aims at the investigation on how the formation of DMSO and subsequent multiphase chemistry affects the $SO_2$ yield, I do not understand why the authors produce DMSO

only in a yield of 0.4 and the residual yield is $SO_2$. The addition pathway has to be revised according to the literature to consider the DMSO formation more adequately including higher yields.

$DMS + OH \rightarrow DMSO + HO_2$    (Schultz et al., 2018)

2) MSIA and MSA are acids. The dissociation of these acids will increase the effective uptake coefficient. In the present study, no dissociation is implemented. Especially, MSA is a strong acid (pKa = -1.92) and should reside predominantly in its dissociated form in aerosol particles and cloud droplets. Hence, as dissociated MSA will not undergo phase transfer, a much higher amount partitions in the aqueous phase as it is calculated by the modelling approach. However, if the pH of aerosols and cloud droplets would be fixed in the model, the effective Henry's Law coefficient can easily be calculated. Then, the implementation of the effective Henry's Law coefficient can be an appropriate way to restrict the numerical costs. Still, in the present model study the pH value is predicted. Thus, the effective Henry's Law coefficient cannot be used. Therefore, the dissociation has to be treated in the model to enable a realistic partitioning and chemistry of MSIA and MSA.

3) The implemented kinetic reaction rate constant for MSA with the OH radical is nearly five times higher than the two other ones given in Barnes et al. (2006). Given the comments on the high rate constant in Barnes et al. (2006), I think the lower ones has to be preferred. This will significantly affect also the oxidation of MSA.

4) The oxidation of DMSO in the gas phase yields only 0.95 MSA. The residual 0.05 should be dimethyl sulfoxide ($DMSO_2$), which will be predominantly being further, but slowly, oxidised into $SO_2$. Thus, to close the mass balance, a 0.05 yield of $SO_2$ should be implemented.

5) The oxidation of DMS can also affect new particle formation. As the authors have stated the formation of MSA by aqueous-phase chemistry will not result in new particle formation. I am missing a discussion how the newly implemented multiphase DMS chemistry scheme affects new particle formation. Have the authors investigated how new particle formation is changed between $R_{all}$ and $R_{std}$? Please address this issue in the discussion of the model results.

**Minor comments**

1) The term 'multiphase' means connection of oxidation in gas and aqueous phase. Therefore, the usage of multiphase mechanism or multiphase oxidation addressing

oxidation in the aqueous phase is wrong. It has to be stated aqueous-phase mechanism or aqueous-phase oxidation.

2) The conversion yield of DMS into $SO_2$ and MSA with the new approach is 78% and 13%, respectively. The addition of these numbers does not reach 100%. What are the residual 9%? Are this DMSO and MSIA? These also needs to be stated.

3) Can the authors please provide how much dry and wet deposition contribute to the lifetime of DMSO, MSIA and MSA?

4) Page 2, Line 25: 'DMS + BrO' and 'DMS + $OH_{(g)}$' spaces are missing

5) Page 3, Line21-22.: Recently, cooking was mentioned as an anthropogenic source for MSA by Dall'Osto et al. (2015)

6) Page 5, Line 8-11: The authors have mentioned that no Cl is produced from the heterogenous reaction 'HOBr + $Cl^-$ + $H^+$' in the model system. This reaction will lead to BrCl formation. Box model studies show that the photolysis of BrCl is an important source for both Cl and Br radicals in the atmosphere (Bräuer et al., 2013), because it recycles HOBr effectively back to Br. As the authors did not include this production pathway for Cl radicals, I wonder if it is implemented for Br radicals? Please clarify this.

7) Page 4, Line 13-14: Can you please provide the resolution of the vertical levels?

8) Page 5, Line 15: 'DMS + $O_{3(aq)}$', 'DMSO + $OH_{(aq)}$', 'MSIA + $OH_{(aq)}$', 'MSIA + $O_{3(aq)}$', 'MSA + $OH_{(aq)}$' spaces are missing

9) Page 5, Line 29-30: Model simulations show that the concentration of OH in the aqueous phase of marine cloud droplets and aerosols can differ up to two orders of magnitude (see Herrmann et al., 2010 or Bräuer et al., 2013). This cannot be derived from the approach of Jacob (2005). Have the authors tried to implement these differences in OH concentration between the bulk of aerosol particles and cloud droplets into the modelling framework?

10) Page 6, Line 7: Usually, these rate constants are measured at 298K not 273K.

11) Page 7, Line 16-17: Measured concentrations of Cl radicals are often in the range of $10^4$ molecules $cm^{-3}$ (Saiz-Lopez and von Glasow, 2012). Have the authors performed a sensitivity study with higher concentrations (e.g. $5x10^3$ molecules $cm^{-3}$)? How is the contribution then affected?

12) Page 9 Line 1: Once MSIA is formed, it is quickly, nearly instantaneous, further oxidised in the atmosphere. If 66% of the precursor DMOS stay below 2 km altitude, which processes trigger the 17% smaller contribution of MSIA?

13) Page 9, Line 8-10: The addition of all percent leads to 99%, please revise to give 100%.

14) Page 9 Line 17: 'Hoffmann' not 'Hoffman'

15) Page 10, Line 13-14: Which OH concentrations are used in these studies? Are these comparable to the present study?

16) Page 11, Line 9-10: Are there no data of DMS available for 2007? What is with the Cape Verde Observatory at which DMS is permanent measured since 2006 (Carpenter et al., 2010)? Why are these data not used?

17) Page 26, Fig. 4: Over Central Asia, the oxidation of DMS by BrO shows the highest source for DMSO. Why is this reaction so strong in this region? Please provide proper reasons. Furthermore, the authors should consider to discuss the modelled concentration patterns of key oxidants.

Overall, I think this paper is appropriate for publication in Atmospheric Chemistry and Physics after successful revision and consideration of the major issues.

**Comments on the formatting of the references**
1) Page 15, Line 19: Formatting error of the DOI number.
2) Page 15, Line 27: Please provide the DOI number.
3) Page 16, Line 3: Formatting error of the DOI number.
4) Page 16, Line 6: Formatting error of the DOI number.
5) Page 16, Line 8: Please provide the DOI number.
6) Page 16, Line 11: Please provide the DOI number.
7) Page 16, Line 26: Please provide the DOI number.
8) Page 17, Line 11: Formatting error of the DOI number.
9) Page 17, Line 30: Formatting error of the DOI number.
10) Page 17, Line 32: Please provide the DOI number.
11) Page 18, Line 2: Please provide the DOI number.
12) Page 18, Line 7: Formatting error of the DOI number.
13) Page 18, Line 9: Please provide the DOI number.
14) Page 18, Line 18: Please provide the DOI number.
15) Page 19, Line 3: Formatting error of the DOI number.
16) Page 19, Line 21: Please provide the DOI number.
17) Page 19, Line 24: Please provide the DOI number.

18) Page 20, Line 29: Formatting error of the DOI number.

19) Page 21, Line 8: Formatting error of the DOI number.

20) Page 21, Line 10: Please provide the DOI number.

21) Page 21, Line 19: Please provide the DOI number.

22) Page 21, Line 25: Formatting error of the DOI number.

23) Page 21, Line 34: Formatting error of the DOI number.

24) Page 22, Line 2: Please provide the DOI number.

25) Page 22, Line 5: Formatting error of the DOI number.

26) Page 22, Line 8: Formatting error of the DOI number.

27) Page 22, Line 12: Formatting error of the DOI number.

28) Page 22, Line 14: Formatting error of the DOI number.

29) Page 22, Line 19: Please provide the DOI number.

30) Page 22, Line 24: Please provide the DOI number.

**References**

Bräuer et al., 2013, Mechanism development and modelling of tropospheric multiphase halogen chemistry: The CAPRAM Halogen Module 2.0 (HM2), J. Atmos. Chem., 70, 19–52.

Carpenter et al., 2010, Seasonal characteristics of tropical marine boundary layer air measured at the Cape Verde Atmospheric Observatory, J. Atmos. Chem. 67, 87–140.

Dall'Osto et al., 2015, On the Origin of AMS "Cooking Organic Aerosol" at a Rural Site, ES&T, 49, 24, 13964-13972.

Herrmann et al., 2010, Tropospheric Aqueous-Phase Free-Radical Chemistry: Radical Sources, Spectra, Reaction Kinetics and Prediction Tools, Chem. Phys. Chem., 11, 3796–3822.

Saiz-Lopez and von Glasow, 2012, Reactive halogen chemistry in the troposphere, Chem. Soc. Rev., 41, 6448–6472.

Schultz et al., 2018, The chemistry–climate model ECHAM6.3-HAM2.3-MOZ1.0, Geosci. Model Dev., 11, 1695–1723.

---

## Referee Comment (RC2) · Anonymous Referee #2 · 13 Jun 2018

This is a good paper that has been well prepared. The goals and methods are clearly presented, and I have mostly textual suggestions that can be found in the attached pdf, along with one or two more serious issues.

I recommend publication when my points are properly addressed.

One main point comes at the end of the paper. The main subject is the importance of halogen chemistry for the DMS –> sulphate oxidation pathway. New intermediates and reactions are included in a global chemistry-transport model.

This gives better validations of intermediates, like MSA and MSA/nssSO4 ratios. Also, the main uncertainties in the reaction pathways are clarified. However, for the formation

of sulphate aerosol the new pathways have limited impact, simply because this is the end-product. Only the speed of the DMS –> sulphate conversion may be affected, and maybe some intermediates have efficient dry- or wet deposition pathways. Thus the phrase "...with a simplified DMS oxidation scheme (gas-phase oxidation by OH and NO3 only) may overestimate sulfate abundances in the pre-industrial environment, potentially leading to underestimates in sulfur aerosol radiative forcing calculations in climate models." seems overstated. I expect no huge impact in sulphate forcing and some reflection on this is recommended.

For the rest of my comments, see annotated manuscript.

Please also note the supplement to this comment:
https://www.atmos-chem-phys-discuss.net/acp-2018-410/acp-2018-410-RC2-supplement.pdf

**Supplement:**

[revised manuscript text omitted]

---

## Author Response (AR1)

**Referee # 1**

We thank Referee #1 for the helpful comments. Please find our responses below.

**General comments**
1) The oxidation of DMS by OH happens by two pathways: H-abstraction from the methyl group and OH-addition onto the sulphur atom. The H-abstraction pathway leads predominantly to SO2. The OH-addition pathway leads predominantly to DMSO (Barnes et al. 2006). The DMSO is then further oxidised into MSIA and subsequently into MSA or SO2 (see von Glasow and Crutzen, 2004; Zhu et al., 2006; Barnes et al. 2006; Hoffmann et al., 2016). A direct conversion of DMS into SO2 is unlikely and only convenient if the OH-addition pathway is parameterised. As the present study aims at the investigation on how the formation of DMSO and subsequent multiphase chemistry affects the SO2 yield, I do not understand why the authors produce DMSO only in a yield of 0.4 and the residual yield is SO2. The addition pathway has to be revised according to the literature to consider the DMSO formation more adequately including higher yields. DMS + OH → DMSO + HO2 (Schultz et al., 2018)

**Response**: As shown in Barnes et al. (2006), the $(CH_3)_2S$-OH adduct is formed via the OH-addition pathway. The $(CH_3)_2S$-OH adduct reacts with $O_2$ to produce mainly DMSO under $NO_x$-free conditions (Arsene et al., 1999; Barnes et al., 2006). However, both Turnipseed et al. (1996) and Hynes et al. (1993) found the yield of DMSO from the $(CH_3)_2S$-OH + $O_2$ + NO reaction to be about 0.5 when NO is present in their experiments. It is difficult to extrapolate the yield of DMSO under atmospheric $NO_x$ conditions and $DMSO_2$ could be formed under low $NO_x$ and low temperature conditions in the remote marine atmosphere (Barnes et al., 2006). The most recent JPL report (Burkholder et al., 2015) also quoted a yield of DMSO from $(CH_3)_2S$-OH + $O_2$ + NO reaction to be 0.5 from Turnipseed et al. (1996) and Hynes et al. (1993).

Pham et al. (1995) uses a yield of 0.6 for $SO_2$ and a yield of 0.4 for DMSO for the addition channel of the DMS+OH reaction in a global three-dimensional chemical transport model, based on laboratory results of Barnes et al. (1988). These yields were also used in other global model studies (Cosme et al., 2002; Spracklen et al., 2005; Breider et al., 2010).

Thus, the OH-addition pathway leads predominantly to DMSO, as argued by the reviewer, only occurs under $NO_x$-free conditions. The reason why Schultz et al. (2018) used a unity yield of DMSO for the additional channel of DMS+OH reaction was not explained in that paper. It seems more common in the modeling community that the yield of DMSO from the addition channel of DMS+OH reaction is not unity. Since the consequence of the addition channel of DMS+OH reaction is still not well understood, we think it is more reasonable to use the same product yields of $SO_2$ (0.6) and DMSO (0.4) as previous modeling studies (Pham et al., 1995; Cosme et al., 2002; Spracklen et al., 2005; Breider et al., 2010). To investigate the importance of this assumption, we perform an additional sensitivity run by using a unity yield of DMSO for the addition channel of the DMS+OH reaction. The global DMSO, MSIA and MSA burden increase

by 33%, 50% and 74%, respectively. The global $SO_2$ and sulfate burden both decrease by 2%.

**Changes in the manuscript**:
- Add $R_{add}$ to Table 3 as a sensitivity run.
- Add MSA/nssSO$_4^{2-}$ ratio for $R_{add}$ run to Fig. 10.
- At Page 5, Line 11: add "Product yields of 0.6 for $SO_2$ and 0.4 for DMSO have been commonly used in global models (Pham et al., 1995; Cosme et al., 2002; Spracklen et al., 2005; Breider et al., 2010) based on experiments described in Turnipseed et al. (1996) and Hynes et al. (1993).".
- At Page 13, Line 19: add "The model run with a unity yield of DMSO from the additional channel of DMS oxidation by OH ($R_{add}$) largely overestimates MSA/nssSO$_4^{2-}$ observations, with $N_{MB}=281\%$.".

2) MSIA and MSA are acids. The dissociation of these acids will increase the effective uptake coefficient. In the present study, no dissociation is implemented. Especially, MSA is a strong acid (pKa = -1.92) and should reside predominantly in its dissociated form in aerosol particles and cloud droplets. Hence, as dissociated MSA will not undergo phase transfer, a much higher amount partitions in the aqueous phase as it is calculated by the modelling approach. However, if the pH of aerosols and cloud droplets would be fixed in the model, the effective Henry's Law coefficient can easily be calculated. Then, the implementation of the effective Henry's Law coefficient can be an appropriate way to restrict the numerical costs. Still, in the present model study the pH value is predicted. Thus, the effective Henry's Law coefficient cannot be used. Therefore, the dissociation has to be treated in the model to enable a realistic partitioning and chemistry of MSIA and MSA.

**Response**: The acid dissociation of MSIA and MSA was considered in the model, although it was not specifically described. We used the same rate constant for the oxidation of MSA$_{(aq)}$ and MS$^-$ by OH$_{(aq)}$, for the oxidation of MSIA$_{(aq)}$ and MSI$^-$ by OH$_{(aq)}$, and for the oxidation of MSIA$_{(aq)}$ and MSI$^-$ by O$_{3(aq)}$. Now we have used different rate constant for each of these reactions in the updated manuscript (Table 1). The updated MSI$^-$+O$_{3(aq)}$ reaction rate constant is about an order of magnitude lower than before, which results in more dissolved MSIA oxidized by OH$_{(aq)}$ than by O$_{3(aq)}$. The updated MS$^-$+OH$_{(aq)}$ reaction rate constant is 4.7 times lower than before, which results in a large increase in the global MSA burden (from about 10 to 20 Gg).

**Changes to the manuscript**:
- In Table 2, add pKa for MSIA and MSA: pKa(MSIA)=2.28 and pKa(MSA)=-1.86 (Hoffmann et al., 2016).
- In Table 1, add "MSI$^-$+OH$_{(aq)}$→MSA$_{(aq)}$", "MSI$^-$+O$_{3(aq)}$→MSA$_{(aq)}$" and "MS$^-$+OH$_{(aq)}$→ SO$_4^{2-}$" and corresponding rate constants.
- At Page 9, Line 21: modified as "Globally, multiphase oxidation in cloud droplets and aerosols by OH$_{(aq)}$ (53%) and O$_{3(aq)}$ (24%) is the biggest sink of MSIA, followed by gas-phase oxidation by OH (19%)."

- At Page 9, Line 30: add "By considering cloud droplets only, Hoffmann (2016) suggested $OH_{(aq)}$ is more important (1.5 times faster) than $O_{3(aq)}$ for MSIA oxidation, which is consistent with our results. Since information such as $OH_{(aq)}$ concentrations in aerosols, aerosol water content and cloud liquid water content were not provided in Hoffmann et al. (2016), we do not further compare our MSIA oxidation by $O_{3(aq)}$ and $OH_{(aq)}$ to Hoffmann et al. (2016)."
- At Page 10, Line 12: change "In $R_{all}$, the global MSA burden is 10 Gg S." into "In $R_{all}$, the global MSA burden is 20 Gg S."
- At Page 10, Line 26: modified as "MSA+$OH_{(aq)}$ accounts for 12% of MSA removal in $R_{all}$, and the rest of MSA is removed via dry (12%) and wet (76%) deposition."
- At Page 12, Line 29: change "Figures 10 and 11 show that modeled MSA/nss$SO_4^{2-}$ ratios are in good agreement with MSA/nss$SO_4^{2-}$ observations" into "Figures 10 and 11 show that modeled MSA/nss$SO_4^{2-}$ ratios calculated from $R_{all}$ can generally reproduce the spatial variability of MSA/nss$SO_4^{2-}$ observations, especially the latitudinal trend of increasing ratios towards the south where anthropogenic sources of nss$SO_4^{2-}$ are less important. However, modeled MSA/nss$SO_4^{2-}$ ratios overestimate observations by a factor of 2 on average"
- More changes in the MSIA and MSA budgets are shown in Section 3.3, 3.4 and 3.6.2 in the updated manuscript.

3) The implemented kinetic reaction rate constant for MSA with the OH radical is nearly five times higher than the two other ones given in Barnes et al. (2006). Given the comments on the high rate constant in Barnes et al. (2006), I think the lower ones has to be preferred. This will significantly affect also the oxidation of MSA.

**Response**: In the updated manuscript, the lower rate constant for MSA oxidation has been used (Table 1) ($1.29 \times 10^7$ $M^{-1}$ $s^{-1}$ at 298 K instead of $6.10 \times 10^7$ $M^{-1}$ $s^{-1}$). The global MSA burden increases from 10 Gg to 20 Gg. The modeled MSA/nss$SO_4^{2-}$ ratio for the run with all new reactions added overestimates the observations by a factor of 2 on average, which suggests more MSA oxidation is needed. The high rate constant for MSA oxidation (4.7 times higher) is now used in the sensitivity run $R_{moreMSA+OH}$.

**Changes to the manuscript**:
- In Table 1, update rate constant for MSA oxidation.
- At Page 10, Line 12: change "In $R_{all}$, the global MSA burden is 10 Gg S." into "In $R_{all}$, the global MSA burden is 20 Gg S."
- At Page 10, Line 26: modified as "MSA+$OH_{(aq)}$ accounts for 12% of MSA removal in $R_{all}$, and the rest of MSA is removed via dry (12%) and wet (76%) deposition."
- At Page 12, Line 29: change "Figures 10 and 11 show that modeled MSA/nss$SO_4^{2-}$ ratios are in good agreement with MSA/nss$SO_4^{2-}$ observations" into "Figures 10 and 11 show that modeled MSA/nss$SO_4^{2-}$ ratios calculated from $R_{all}$ can generally reproduce the spatial variability of MSA/nss$SO_4^{2-}$ observations, especially the latitudinal trend of increasing ratios towards the south where anthropogenic sources of nss$SO_4^{2-}$ are less important. However, modeled MSA/nss$SO_4^{2-}$ ratios overestimate observations by a factor of 2 on average"
- At Page 13, Line 15: add "Due to the small chemical loss of MSA in our model,

MSA/nssSO$_4^{2-}$ in model run without MSA + OH$_{(aq)}$ ($R_{noMSA+OH(aq)}$) is similar to that in $R_{all}$. The model run with a larger reaction rate coefficient of MSA + OH$_{(aq)}$ ($R_{moreMSA+OH(aq)}$) results in a decrease in modeled MSA/nssSO$_4^{2-}$ (24% on average) compared to $R_{all}$."

4) The oxidation of DMSO in the gas phase yields only 0.95 MSA. The residual 0.05 should be dimethyl sulfoxide (DMSO2), which will be predominantly being further, but slowly, oxidised into SO2. Thus, to close the mass balance, a 0.05 yield of SO2 should be implemented.

**Response**: In the updated manuscript, we have applied a 0.05 yield of SO2 to the oxidation of DMSO in the gas phase. This results in negligible changes in the SO$_2$ budget, as it is a very small source of SO$_2$ (Fig. 1).

**Changes to the manuscript**:
- Update "DMSO+OH→ 0.95MSIA+0.05SO$_2$$^{(new)}$" in Table 1.
- Add a pathway "DMSO→SO$_2$" with 98 Gg S yr$^{-1}$ in Fig. 1.

5) The oxidation of DMS can also affect new particle formation. As the authors have stated the formation of MSA by aqueous-phase chemistry will not result in new particle formation. I am missing a discussion how the newly implemented multiphase DMS chemistry scheme affects new particle formation. Have the authors investigated how new particle formation is changed between Rall and Rstd? Please address this issue in the discussion of the model results.

**Response**: We did not investigate how new particle formation is changed between $R_{all}$ and $R_{std}$. This will be an important investigation in the future using an aerosol microphysics model simulation.

**Changes to the manuscript**:
- At Page 14, Line 16: add "Quantifying the impacts of our updated sulfur oxidation scheme on new particle formation is out of the scope of this study and should be addressed in the future.".

Minor comments
1) The term 'multiphase' means connection of oxidation in gas and aqueous phase. Therefore, the usage of multiphase mechanism or multiphase oxidation addressing oxidation in the aqueous phase is wrong. It has to be stated aqueous-phase mechanism or aqueous-phase oxidation.

**Response:** In the updated manuscript, we use multiphase oxidation for the process involving gas and aqueous phase, and we use aqueous-phase oxidation for the process happening in the liquid.

**Changes to the manuscript**:
- In Table 1 and Table 4, change all multiphase reactions into aqueous-phase reactions with corresponding rate coefficients.

2) The conversion yield of DMS into SO2 and MSA with the new approach is 78% and 13%, respectively. The addition of these numbers does not reach 100%. What are the residual 9%? Are this DMSO and MSIA? These also needs to be stated.

**Response:** The conversion yield of DMS into $SO_2$ and MSA in the new approach should not be 100% due to loss of the intermediates DMSO and MSIA via dry and wet deposition.

**Changes to the manuscript**:
- At Page 14, Line1: modified as "Compared to the standard GEOS-Chem model run, the updated sulfur scheme in this study decreases the conversion yield of DMS to $SO_2$ ($Y_{DMS \rightarrow SO2}$) from 91% to 75% and increases the conversion yield of DMS to MSA ($Y_{DMS \rightarrow MSA}$) from 9% to 15%. The remaining 10% of DMS is lost via wet and dry deposition of DMSO and MSIA."
- In the abstract, add "The remaining 10% of DMS is lost via deposition of intermediates DMSO and MSIA."

3) Can the authors please provide how much dry and wet deposition contribute to the lifetime of DMSO, MSIA and MSA?

**Response**: The dry deposition loss contributes 16%, 2% and 12% to the lifetime of DMSO, MSIA and MSA, respectively. The wet deposition loss contributes 11%, 2% and 76% to the lifetime of DMSO, MSIA and MSA, respectively. Chemical oxidation is responsible for the rest of the losses.

**Changes to the manuscript**:
- At Page 9, Line 1: modified as "DMSO is removed from the atmosphere via gas-phase oxidation by OH (33%), multiphase oxidation by OH in cloud droplets (37%) and aerosols (3%), and dry (16%) and wet deposition (11%). The lifetime of DMSO is about 11 hours.".
- At Page 9, Line 20: modified as "MSIA is mainly removed in the troposphere via both gas-phase and multiphase oxidation by OH, with a lifetime of 4 hours. Dry (2%) and wet (2%) deposition of MSIA accounts for 4% of MSIA removal in the troposphere. Globally, multiphase oxidation in cloud droplets and aerosols by $OH_{(aq)}$ (53%) and $O_{3(aq)}$ (24%) is the biggest sink of MSIA, followed by gas-phase oxidation by OH (19%)."
- At Page 10, Line 19: modified as "MSA+$OH_{(aq)}$ accounts for 12% of MSA removal in $R_{all}$, and the rest of MSA is removed via dry (12%) and wet (76%) deposition. The lifetime of MSA is 2.2 days globally".

4) Page 2, Line 25: 'DMS + BrO' and 'DMS + OH(g)' spaces are missing

**Response**: We have updated the manuscript accordingly.

**Changes to the manuscript**:
- Add spaces.

5) Page 3, Line21-22.: Recently, cooking was mentioned as an anthropogenic source for MSA by Dall'Osto et al. (2015)

**Response**: We have updated the manuscript accordingly.

**Changes to the manuscript**:
- At Page 3, Line 23: change "MSA has only a natural source" into "MSA is generally considered to have a predominant natural origin".

6) Page 5, Line 8-11: The authors have mentioned that no Cl is produced from the heterogenous reaction 'HOBr + Cl- + H+' in the model system. This reaction will lead to BrCl formation. Box model studies show that the photolysis of BrCl is an important source for both Cl and Br radicals in the atmosphere (Bräuer et al., 2013), because it recycles HOBr effectively back to Br. As the authors did not include this production pathway for Cl radicals, I wonder if it is implemented for Br radicals? Please clarify this.

**Response:** In our model, we used monthly mean Cl mixing ratios from Sherwen et al. (2016). The reason why we used their Cl mixing ratio outputs in our model is that they have a more detailed chlorine chemistry scheme (considering Cl-Br-I coupling) than ours. They considered "$HOBr+Cl^-+H^+ \rightarrow BrCl$" in cloud droplets and sulfate aerosols in their model, but did not include this reaction on sea salt aerosols as this led to unrealistically high bromine abundances over the ocean.

Our model is based on Chen et al. (2017), which includes "$HOBr+Cl^-/Br^-+H^+ \rightarrow BrCl + Br_2$" reaction in both cloud droplets, sulfate aerosols and sea salt aerosols. We consider inclusion of sea salt debromination to be a better parameterization of bromine budget in the marine boundary layer. Thus, the BrO mixing ratios in our model are the BrO mixing ratios in Chen et al. (2017) that included "$HOBr+Cl^-+H^+ \rightarrow BrCl$". In sum, in our model Cl mixing ratios are from Sherwen et al. (2016) and BrO mixing ratios are from Chen et al. (2017).

**Changes to the manuscript**:
- At Page 5, Line 13: Delete "from the $HOBr+Cl^-+H^+$" reaction" from the sentence "We used monthly mean Cl mixing ratios from Sherwen et al. (2016), which considered Cl-Br-I coupling but did not include chlorine production from the $HOBr+Cl^-+H^+$" reaction on sea salt aerosols that was suggested to be the largest tropospheric chlorine source in Schmidt et al. (2016)."

7) Page 4, Line 13-14: Can you please provide the resolution of the vertical levels?

**Response**: The matrix below shows the altitude of the upper bound of each layer in km. For example, the first layer is from 0 to 0.123 km; the second layer is from 0.123 km to 0.254 km; the 47th is from 68.392 km to 80.581 km.

[0.123 0.254  0.387  0.521  0.657  0.795  0.934  1.075  1.218  1.363  1.510  1.659
1.860  2.118  2.382  2.654  2.932  3.219  3.665  4.132  4.623  5.142  5.692  6.277
6.905  7.582  8.320  9.409  10.504 11.578 12.633 13.674 14.706 15.731 16.753 17.773
19.855 22.004 24.24  26.596 31.716 37.574 44.286 51.788 59.924 68.392 80.581]

**Changes to the manuscript**:
- At Page 4, Line 14: Add "The vertical layer thickness ranges from 120-150 m for the first 12 layers to 200-800 m for the 13th-27th layers and >1000 m for the rest (http://acmg.seas.harvard.edu/geos/doc/archive/man.v9-01-02/appendix_3.html#A3.5.2)." .

8) Page 5, Line 15: 'DMS + O3(aq)', 'DMSO + OH(aq)', 'MSIA + OH(aq)', 'MSIA + O3(aq)', 'MSA + OH(aq)' spaces are missing

**Response**: We have updated the manuscript accordingly.

**Changes to the manuscript**: Add spaces.

9) Page 5, Line 29-30: Model simulations show that the concentration of OH in the aqueous phase of marine cloud droplets and aerosols can differ up to two orders of magnitude (see Herrmann et al., 2010 or Bräuer et al., 2013). This cannot be derived from the approach of Jacob (2005). Have the authors tried to implement these differences in OH concentration between the bulk of aerosol particles and cloud droplets into the modelling framework?

**Response:** We did not implement different OH concentrations between the bulk aerosol particles and cloud droplets in the model. Instead, we did one sensitivity run by reducing OH concentrations in both cloud droplets and aerosols by two orders of magnitude ($R_{lowOH(aq)}$). Following the reviewer's suggestion, we run a sensitivity run by reducing the aerosol OH concentration by a factor of 20, following the maritime condition in Herrmann et al. (2010). As expected, the global DMSO and MSIA burden increase by 1% and MSA burden decreases by 2%. The small changes are due to the fact that the aqueous-phase oxidation by $OH_{(aq)}$ occurs mainly in clouds instead of aerosols in our model and less oxidation by $OH_{(aq)}$ in aerosols is compensated by more oxidation in other forms (e.g. oxidation in clouds).

**Changes to the manuscript**:
- At Page 6, Line 11: add "We conduct another sensitivity simulation by reducing the $[OH_{(aq)}]$ in aerosols only by a factor of 20 (Herrmann et al., 2010) and found negligible

changes (<2%) in the global sulfur burden.".

10) Page 6, Line 7: Usually, these rate constants are measured at 298K not 273K.

**Response:** In the updated manuscript, we do not have this temperature limitation. More aqueous-phase oxidation can occur at low temperature, which affects especially the MSIA burden as aqueous-phase oxidation is the main sink of MSIA (Fig. 1). We agree this is a more reasonable parameterization.

**Changes to the manuscript**:
- At Page 6, Line 16: delete "Multiphase sulfur reactions added in the model are only activated when the air temperature is above 273 K, to be consistent with the temperature at which their rate constants were obtained (Table 1).".

11) Page 7, Line 16-17: Measured concentrations of Cl radicals are often in the range of 104 molecules cm-3 (Saiz-Lopez and von Glasow, 2012). Have the authors performed a sensitivity study with higher concentrations (e.g. 5x103 molecules cm-3)? How is the contribution then affected?

**Response:** In the updated manuscript, we have conducted a sensitivity run by increasing the Cl mixing ratio by an order of magnitude. The fraction of DMS oxidized by Cl increases from 4% to 28%. This results in changes in the global DMS (-29%), DMSO (-2%), MSIA (-12%), MSA (10%), $SO_2$ (-2%) and sulfate (-3%) burden.

**Changes to the manuscript**:
- At Page 7, Line 17: add "$f_{[l]DMS+Cl}$ increases to 28% in a sensitivity run increasing Cl mixing ratios by an order of magnitude.".
- Ag Page 8, Line 2: add "However, a recent study suggests that this is an overestimate of tropospheric Cl abundance (Gromov et al., 2018)."

12) Page 9 Line 1: Once MSIA is formed, it is quickly, nearly instantaneous, further oxidised in the atmosphere. If 66% of the precursor DMOS stay below 2 km altitude, which processes trigger the 17% smaller contribution of MSIA?

**Response:** The reason why a larger fraction of DMSO stays below 2 km than MSIA is that MSIA is oxidized faster than DMSO in the marine boundary layer. As shown in Table 1, the reaction rate constant for $DMSO_{(aq)}+OH_{(aq)}$ reaction is $6.63\times10^9$ $M^{-1}$ $s^{-1}$ at 298 K while the reaction rate constant for $MSI^-+OH_{(aq)}$ reaction is $1.2\times10^{10}$ $M^{-1}$ $s^{-1}$ in the model. In addition, MSIA can also be oxidized by $O_{3(aq)}$, which results in even shorter lifetime of MSIA in the marine boundary layer (below 2 km). The MSIA produced via gas-phase oxidation by OH in the upper troposphere has a longer lifetime compared to MSIA in the marine boundary layer due to smaller amount of clouds and aerosols for the MSIA oxidation in the aqueous phase in the upper troposphere. Thus, a larger amount of

MSIA stays in the upper troposphere (above 2 km).

**Changes to the manuscript**:
- At Page 9, Line 4: add "The smaller fraction of MSIA below 2 km compared to DMSO is due to faster oxidation of MSIA by $OH_{(aq)}$ and $O_{3(aq)}$ in clouds and aerosols (Table 1)."

13) Page 9, Line 8-10: The addition of all percent leads to 99%, please revise to give 100%.

**Response**: We have updated this accordingly.

**Changes to the manuscript**:
- At Page 9, Line 10, modified as "MSIA is mainly removed in the troposphere via both gas-phase and multiphase oxidation by OH, with a lifetime of 4 hours. Dry (2%) and wet (2%) deposition of MSIA accounts for 4% of MSIA removal in the troposphere. Globally, multiphase oxidation in cloud droplets and aerosols by $OH_{(aq)}$ (53%) and $O_{3(aq)}$ (24%) is the biggest sink of MSIA, followed by gas-phase oxidation by OH (19%).".

14) Page 9 Line 17: 'Hoffmann' not 'Hoffman'

**Response**: Thank you, this has been fixed.

**Changes to the manuscript**: Change "Hoffman" into "Hoffmann".

15) Page 10, Line 13-14: Which OH concentrations are used in these studies? Are these comparable to the present study?

**Response**: None of those studies (Pham et al, 1995; Chin et al., 1996; 2000; Cosme et al., 2002; Hezel et al., 2011) reported global OH concentrations. In addition, none of those studies included oxidation of MSA by $OH_{(aq)}$ in clouds and aerosols, as mentioned in the manuscript. The OH concentration in our model has a global annual mean of $1.3 \times 10^6$ molecule $cm^{-3}$.

**Changes to the manuscript**:
- At Page 10, Line 22: add "Information about the global distribution of MSA concentrations and deposition from these previous modelling studies are needed for comparison."

16) Page 11, Line 9-10: Are there no data of DMS available for 2007? What is with the Cape Verde Observatory at which DMS is permanent measured since 2006 (Carpenter et al., 2010)? Why are these data not used?

**Response**: We are not able to find reported DMS data for 2007. We would like to

compare our model results with other DMS observations once provided.

**Changes to the manuscript**: No changes.

17) Page 26, Fig. 4: Over Central Asia, the oxidation of DMS by BrO shows the highest source for DMSO. Why is this reaction so strong in this region? Please provide proper reasons. Furthermore, the authors should consider to discuss the modeled concentration patterns of key oxidants.

**Response**: The oxidation of DMS by BrO shows the highest source for DMSO over central Asia is because high BrO abundance predicted in the model due to less reactive bromine deposition there (Schmidt et al., 2016). It should be noted that both DMS and DMSO are at very low concentration over central Asia as DMS is emitted from the ocean. This study focus on sulfur oxidation over the ocean. In the updated manuscript, we have added a global distribution of BrO, Cl, OH and $O_3$ in Figure 12. This study focus on sulfur oxidation and we refer to previous literatures for reasons causing the global distribution of these oxidants.

**Changes to the manuscript**:
- Add Fig. 12.
- At Page 5, Line 15: add "The global distributions of tropospheric annual-mean concentrations of BrO, Cl, OH and $O_3$ are shown in Fig. 12. The high BrO abundances over subtropics and polar regions are due to low deposition fluxes of reactive bromine (Schmidt et al., 2016) and the high BrO abundance over Southern Ocean is due to its source from sea salt debromination (Chen et al., 2017). The high Cl abundance over coastal regions in the Northern Hemisphere is due to heterogeneous uptake of $N_2O_5$ on sea salt aerosols to produce reactive chlorine (Sherwen et al., 2017)."

(18) Comments on the formatting of the references
1) Page 15, Line 19: Formatting error of the DOI number.
2) Page 15, Line 27: Please provide the DOI number.
3) Page 16, Line 3: Formatting error of the DOI number.
4) Page 16, Line 6: Formatting error of the DOI number.
5) Page 16, Line 8: Please provide the DOI number.
6) Page 16, Line 11: Please provide the DOI number.
7) Page 16, Line 26: Please provide the DOI number.
8) Page 17, Line 11: Formatting error of the DOI number.
9) Page 17, Line 30: Formatting error of the DOI number.
10) Page 17, Line 32: Please provide the DOI number.
11) Page 18, Line 2: Please provide the DOI number.
12) Page 18, Line 7: Formatting error of the DOI number.
13) Page 18, Line 9: Please provide the DOI number.
14) Page 18, Line 18: Please provide the DOI number.
15) Page 19, Line 3: Formatting error of the DOI number.

16) Page 19, Line 21: Please provide the DOI number.
17) Page 19, Line 24: Please provide the DOI number.
18) Page 20, Line 29: Formatting error of the DOI number.
19) Page 21, Line 8: Formatting error of the DOI number.
20) Page 21, Line 10: Please provide the DOI number.
21) Page 21, Line 19: Please provide the DOI number.
22) Page 21, Line 25: Formatting error of the DOI number.
23) Page 21, Line 34: Formatting error of the DOI number.
24) Page 22, Line 2: Please provide the DOI number.
25) Page 22, Line 5: Formatting error of the DOI number.
26) Page 22, Line 8: Formatting error of the DOI number.
27) Page 22, Line 12: Formatting error of the DOI number.
28) Page 22, Line 14: Formatting error of the DOI number.
29) Page 22, Line 19: Please provide the DOI number.
30) Page 22, Line 24: Please provide the DOI number.

**Response**: Thanks for the comments. We have updated the manuscript accordingly.

**Changes to the manuscript**:
- Format the references accordingly.

**Referee # 2**

We thank Referee #2 for the helpful comments. Please find our responses below.

One main point comes at the end of the paper. The main subject is the importance of halogen chemistry for the DMS –> sulphate oxidation pathway. New intermediates and reactions are included in a global chemistry-transport model. This gives better validations of intermediates, like MSA and MSA/nssSO4 ratios. Also, the main uncertainties in the reaction pathways are clarified. However, for the formation of sulphate aerosol the new pathways have limited impact, simply because this is the end-product. Only the speed of the DMS –> sulphate conversion may be affected, and maybe some intermediates have efficient dry- or wet deposition pathways. Thus the phrase "...with a simplified DMS oxidation scheme (gas-phase oxidation by OH and NO3 only) may overestimate sulfate abundances in the pre-industrial environment, potentially leading to underestimates in sulfur aerosol radiative forcing calculations in climate models." seems overstated. I expect no huge impact in sulphate forcing and some reflection on this is recommended. For the rest of my comments, see annotated manuscript. Please also note the supplement to this comment: https://www.atmos-chem-phys-discuss.net/acp-2018-410/acp-2018-410-RC2- supplement.pdf

**Response:** We agree that the new pathways added in model could have limited impact on the sulfate burden if the depositions of DMSO, MSIA and MSA are small so that DMS will be oxidized to form sulfate as an end-product. However, if a large amount of MSA produced from DMS oxidation is removed via deposition instead of being further oxidized to form sulfate, then it could potentially result in a non-negligible change in the sulfate burden. This is the case in this study when using a lower rate coefficient (updated) for the oxidation of MSA by $OH_{(aq)}$ in cloud droplets and aerosols. We think it is still necessary to mention that the updated sulfur chemistry scheme **may** be important for sulfur aerosol radiative forcing calculations in climate models.

**Changes to the manuscript:**
- At Page 14, Line 1: modified as "Compared to the standard GEOS-Chem model run, the updated sulfur scheme in this study decreases the conversion yield of DMS to $SO_2$ ($Y_{DMS \rightarrow SO2}$) from 91% to 75% and increases the conversion yield of DMS to MSA ($Y_{DMS \rightarrow MSA}$) from 9% to 15%. The remaining 10% of DMS is lost via wet and dry deposition of DMSO and MSIA. In order to gain insight into the impacts of our updated sulfur scheme on global $SO_2$, MSA and sulfate burden, we conducted two sensitivity studies by allowing DMS as the only sulfur source for both the standard model run $R_{std}$ ($R_{std\_onlyDMS}$) and full model run $R_{all}$ ($R_{all\_onlyDMS}$). Compared to $R_{std\_onlyDMS}$, the global DMS, $SO_2$, MSA and sulfate burden in $R_{all\_onlyDMS}$ decreases by 40%, 17%, 8% and 12%, respectively. The decrease in DMS is mainly due to DMS oxidation by BrO with the updated sulfur scheme. The decrease in $SO_2$ is due to a lower yield of $SO_2$ from DMS ($Y_{DMS \rightarrow SO2}$), but is partly compensated by the increase in the DMS oxidation rate. MSA decreases despite an increase in the yield of MSA from DMS ($Y_{DMS \rightarrow MSA}$) due to a shorter lifetime in $R_{all\_onlyDMS}$ (2.2 days in $R_{all\_onlyDMS}$ versus 4.1 days in $R_{std\_onlyDMS}$) that is caused by the aqueous-phase sink of MSA via MSA + $OH_{(aq)}$ and faster deposition of

MSA produced in the MBL. The decrease in sulfate is caused by the decrease in $SO_2$ but is partly compensated by the inclusion of MSA + $OH_{(aq)}$ as a sulfate source, which accounts for 4% of global sulfate production. The decrease in sulfate will be smaller if more MSA is oxidized into sulfate instead of being lost via deposition. In sum, climate models with a simplified DMS oxidation scheme (gas-phase oxidation by OH and $NO_3$ only) may overestimate $SO_2$, MSA and sulfate abundances in the pre-industrial environment, potentially leading to underestimates in sulfur aerosol radiative forcing calculations in climate models. Quantifying the impacts of our updated sulfur oxidation scheme on new particle formation is out of the scope of this study and should be addressed in the future."

Other comments from the second reviewer shown in the supplement have been accepted in the updated manuscript.

(1) **Comment:** Page 8, Line 7 "The difference between the results from Boucher et al. (2003) and this study is that Boucher et al. (2003) did not include DMS+BrO in their model simulation, 5 which could lead to an overestimate of the contribution of DMS+$O_{3(aq)}$ as both reactions are most important over the high-latitude oceans during winter.": since this channel only accounts for 12% of the DMS oxidation, I guess this argument is weak. I would expect that this "missing" oxidation would be divided equally over the remaining DMS oxidation channels.

**Response:** We agree with the reviewer that the smaller contribution of $O_{3(aq)}$ to DMS oxidation in our study (2%) compared to Boucher et al. (2003) (6%) is not solely caused by the inclusion of DMS+BrO in our study. The fraction of DMS oxidized by O3(aq) increases from 1.8% in $R_{all}$ to only 2.4% in a sensitivity run without DMS+BrO reaction ($R_{noDMS+BrO}$). We have updated the manuscript accordingly.

**Changes to the manuscript:**
- Page 8, Line 16, modified as "The difference between the results from Boucher et al. (2003) and this study could be due to the differences in oxidants abundances such as $O_3$, OH, BrO and Cl."

(2) **Comment:** Page 8, Line 27 "Multiphase oxidation is especially important over regions where clouds are frequent and OH concentrations are high, e.g. low- to mid-latitude oceans (Fig. 5).": somehow counterintuitive, since the gas-phase OH reaction pathway is more important than the multiphase oxidation.

**Response:** In our model, DMSO is removed from the atmosphere via deposition and oxidation by OH in both the gas phase and the aqueous phase. The aqueous-phase oxidation occurs faster than gas-phase oxidation when a lot of clouds are present (low- to mid-latitude oceans).

**Changes to the manuscript:**

- Page 9, Line 4: modified as "Multiphase oxidation mainly occurs over regions where clouds are frequent and OH concentrations are high, e.g. low- to mid-latitude oceans (Fig. 5)."
- Page 9, Line 17: modified as "Multiphase production of MSIA mainly occurs over low- to mid-latitude oceans where the OH abundance is high and clouds are frequent (Fig. 6)."

(3) **Comment:** Page 12, Line 16: Add the 23 stations description to a table.

**Response:** The manuscript is updated accordingly.

**Changes to the manuscript:**

[revised manuscript text omitted]

Qianjie Chen 7/14/18 4:15 PM

Qianjie Chen 7/14/18 10:41 AM

Qianjie Chen 7/14/18 10:41 AM

Qianjie Chen 7/14/18 10:41 AM

Qianjie Chen 7/14/18 10:41 AM

Qianjie Chen 7/14/18 10:02 AM

Qianjie Chen 7/14/18 10:02 AM

Qianjie Chen 7/14/18 10:02 AM

Qianjie Chen 7/14/18 10:41 AM

Qianjie Chen 7/14/18 10:41 AM

Qianjie Chen 7/14/18 10:15 AM

Qianjie Chen 7/14/18 10:21 AM

Qianjie Chen 7/14/18 10:17 AM

| | | | |
|---|---|---|---|
| $HSO_3^- + H_2O_{2(aq)} + H^+ \rightarrow SO_4^{2-} + 2H^+ + H_2O_{(aq)}$ | $2.36\times10^{3(c)}$ | -4760 | Jacob (1986) |
| $HSO_3^- + O_{3(aq)} \rightarrow SO_4^{2-} + H^+ + O_{2(aq)}$ | $3.20\times10^5$ | -4830 | Jacob (1986) |
| $SO_3^{2-} + O_{3(aq)} \rightarrow SO_4^{2-} + O_{2(aq)}$ | $1.00\times10^9$ | -4030 | Jacob (1986) |
| $S(IV) + O_{2(aq)} \xrightarrow{Mn(II),Fe(III)} SO_4^{2-}$ | see note[d] | | Martin and Good (1991) |
| $HSO_3^- + HOBr_{(aq)} \rightarrow SO_4^{2-} + 2H^+ + Br^-$ | $3.20\times10^9$ | 0 | Liu(2000);Chen et al.(2016; 2017) |
| $SO_3^{2-} + HOBr_{(aq)} \rightarrow SO_4^{2-} + H^+ + Br^-$ | $5.00\times10^9$ | 0 | Troy and Margerum (1991) |

[new] New reaction added in the model.

[a] $k(T, [O_2], [M]) = 8.2\times10^{-39}[O_2]e^{5376/T}/(1+1.05\times10^{-5}([O_2]/[M])e^{3644/T})$ cm$^3$ molecule$^{-1}$ s$^{-1}$.

[b] low pressure limit: $3.3\times10^{-31}(300/T)^{4.3}$ cm$^6$ molecule$^{-2}$ s$^{-1}$; high pressure limit: $1.6\times10^{-12}$ cm$^3$ molecule$^{-1}$ s$^{-1}$.

[c] Rate constant between $HSO_3^- + H_2O_{2(aq)}$ at pH=4.5.

5    [d] The metal-catalyzed sulfate production rate is calculated from the following expression:

$$-\frac{d[SO_4^{2-}]}{dt} = 750[Mn(II)][S(IV)] + 2600[Fe(III)][S(IV)] + 1.0\times10^{10}[Mn(II)][Fe(III)][S(IV)]$$

Detailed description about [Mn(II)] and [Fe(III)] concentrations can be found in Alexander et al. (2009).

[e] IUPAC: http://iupac.pole-ether.fr/htdocs/datasheets/pdf/SOx13_Cl_CH3SCH3.pdf

10    **Table 2.** Henry's law constant at 298 K ($H_{X(298)}$), mass accommodation coefficient ($\alpha_b$) and aqueous-phase diffusivity at 298 K ($D_{l(298K)}$) for DMS, DMSO, MSIA and MSA, and acid dissociation constant (pK$_a$) for MSIA and MSA at 298 K.

| | $H_{X(298)}$ [M atm$^{-1}$] | $-\Delta H/R$ [K] | Reference | pK$_a$ | Reference | $\alpha_b$ | Reference | $D_{l(298K)}$ [m$^2$ s$^{-1}$] | Reference |
|---|---|---|---|---|---|---|---|---|---|
| DMS | 0.56 | -4480 | Campolongo et al. (1999) | / | / | 0.001 | Zhu et al. (2006) | $1.5\times10^{-5}$ | Saltzman et al. (1993) |
| DMSO | $1\times10^7$ | -2580 | Campolongo et al. (1999) | / | / | 0.1 | Zhu et al. (2006) | $1.0\times10^{-5}$ | Zhu et al. (2003) |
| MSIA | $1\times10^8$ | -1760 | Campolongo et al. (1999) | 2.28[a] | Wudl et al. (1967) | 0.1 | Zhu et al. (2006) | $1.2\times10^{-5}$ | Same as MSA |
| MSA | $1\times10^9$ | -1760 | Campolongo et al. (1999) | -1.86[b] | Clarke and Woodward (1966) | 0.1 | Zhu et al. (2006) | $1.2\times10^{-5}$ | Schweitzer et al. (1998) |

[a] $CH_3SO_2H \leftrightarrow CH_3SO_2^- + H^+$

[b] $CH_3SO_3H \leftrightarrow CH_3SO_3^- + H^+$

**Table 3.** Overview of model runs.

| Model run | Specification |
|---|---|
| $R_{\mathrm{all}}$ | Full model run including all reactions described in Table 1, including the DMSO and MSIA intermediates; sea surface water DMS concentration obtained from Lana et al. (2011) |
| $R_{\mathrm{std}}$ | Standard run which includes gas-phase oxidation of DMS by OH and $NO_3$ only, with no DMSO or MSIA intermediates |
| $R_{\mathrm{Kettle}}$ | $R_{\mathrm{all}}$; sea surface water DMS concentration obtained from Kettle et al. (1999) |
| $R_{\mathrm{noDMS+BrO}}$ | $R_{\mathrm{all}}$; without DMS+BrO reaction |
| $R_{\mathrm{noMUL}}$ | $R_{\mathrm{all}}$; without multiphase oxidation of DMS, DMSO, MSIA and MSA |
| $R_{\mathrm{noMSA+OH(aq)}}$ | $R_{\mathrm{all}}$; without MSA+OH$_{(aq)}$ reaction |
| $R_{\mathrm{lessMSA+OH(aq)}}$ | $R_{\mathrm{all}}$; $k_{\mathrm{MSA+OH(aq)}}$/4.7 (Zhu et al., 2003) |
| $R_{\mathrm{lowOH(aq)}}$ | $R_{\mathrm{all}}$; reduce OH$_{(aq)}$ concentrations in cloud droplets and aerosols by a factor of 100 |
| $R_{\mathrm{add}}$ | $R_{\mathrm{all}}$; a unity yield of DMSO for the addition channel of DMS+OH reaction[a] |
| $R_{\mathrm{10Cl}}$ | $R_{\mathrm{10Cl}}$; increase Cl mixing ratios by a factor of 10 |
| $R_{\mathrm{all\_onlyDMS}}$ | $R_{\mathrm{all}}$; DMS emission from the ocean is the only sulfur source |
| $R_{\mathrm{std\_onlyDMS}}$ | $R_{std}$; DMS emission from the ocean is the only sulfur source |

[a]The product yield for the addition channel of the DMS+OH reaction is highly uncertain. Product yields of 0.6 for $SO_2$ and 0.4 for DMSO have been commonly used in global models (Pham et al., 1995; Cosme et al., 2002; Spracklen et al., 2005; Breider et al., 2010) based on experiments described in Turnipseed et al. (1996) and Hynes et al. (1993), and is used in this study (e.g., in $R_{\mathrm{all}}$). Experiments under $NO_x$-free conditions suggest a DMSO yield near unity (Arsene et al., 1999; Barnes et al., 2006), as used in the sensitivity simulation $R_{\mathrm{add}}$.

Qianjie Chen 8/1/18 4:20 PM

**Table 4.** The uncertainties of the rate constants for the 12 reactions added in the model. The uncertainty factor $f_{298}$ means the reaction rate constant may be greater than or less than the recommended value by the factor $f_{298}$. Type "R", "L" and "M" represents values obtained from " literature reviews", "laboratory measurements" and "modeling studies", respectively.

| Gas-phase reactions | $f_{298}$ | Type | Reference |
|---|---|---|---|
| DMS+OH $\xrightarrow{addition}$ ... | 1.2 | R | Burkholder et al. (2015) |
| DMS+BrO → ... | 1.3 | R | Burkholder et al. (2015) |
| DMS+O$_3$ → ... | 1.2 | L | Du et al. (2007) |
| DMS+Cl → ... | 1.2 | R | Burkholder et al. (2015) |
| DMSO+OH→ ... | 1.2 | R | Burkholder et al. (2015) |
| MSIA+OH→ ... | 1.4 | R | Burkholder et al. (2015) |
| MSIA+O$_3$→ ... | 1.5 | M | Lucas and Prinn (2002) |
| **Aqueous-phase reactions** | $k_{298}$ [M$^{1-n}$ s$^{-1}$] | Type | Reference |
| DMS$_{(aq)}$+O$_{3(aq)}$ → ... | $(8.6\pm8.1)\times10^8$ | L | Gershenzon et al. (2001) |
|  | $(6.1\pm2.4)\times10^8$ | L | Lee and Zhou (1994) |
| DMSO$_{(g)}$+OH$_{(aq)}$ → ... | $(6.6\pm0.7)\times10^9$ | L | Zhu et al. (2003) |
|  | $7.5\times10^9$ | M | Hoffmann et al. (2016) |
|  | $(4.5\pm0.4)\times10^9$ | L | Bardouki et al. (2002) |
|  | $(5.4\pm0.3)\times10^9$ | L | Milne et al. (1989) |
| MSIA$_{(aq)}$+OH$_{(aq)}$→ ... | $(6.0\pm1.0)\times10^9$ | L | Sehested and Holcman (1996) |
| MSI$^-$+OH$_{(aq)}$→ ... | $(1.2\pm0.2)\times10^{10}$ | L | Bardouki et al. (2002) |
|  | $7.7\times10^9$ | M | Zhu et al. (2006) |
| MSIA+O$_{3(aq)}$→ ... | $3.5\times10^7$ | M | Hoffmann et al. (2016) |
| MSI$^-$+O$_{3(aq)}$ → ... | $2.0\times10^6$ | L | Flyunt et al. (2001) |
| MSA$_{(aq)}$+OH$_{(aq)}$ → ... | $1.5\times10^7$ | M | Hoffmann et al. (2016) |
| MS$^-$+OH$_{(aq)}$ → ... | $(1.3\pm0.1)\times10^7$ | L | Zhu et al. (2003) |
|  | $(6.1\pm1.1)\times10^7$ | L | Milne et al. (1989) |

Qianjie Chen 7/15/18 3:36 PM

Qianjie Chen 7/15/18 3:44 PM

Qianjie Chen 7/15/18 3:41 PM

Qianjie Chen 7/15/18 3:36 PM
**Formatted Table**

Qianjie Chen 7/15/18 3:44 PM

Qianjie Chen 7/15/18 3:44 PM

Qianjie Chen 7/15/18 3:44 PM

Qianjie Chen 7/15/18 3:45 PM

Qianjie Chen 7/15/18 3:47 PM

Qianjie Chen 7/15/18 3:47 PM

Qianjie Chen 7/15/18 3:50 PM

Qianjie Chen 7/15/18 3:50 PM

Qianjie Chen 7/15/18 4:07 PM

Qianjie Chen 7/15/18 4:04 PM

Qianjie Chen 7/15/18 4:07 PM

Qianjie Chen 7/15/18 4:07 PM

Qianjie Chen 7/15/18 4:07 PM

**Table 5.** The locations of the 23 stations that provide annual-mean MSA/nssSO$_4^{2-}$ observations.

| Station name | Location | Station name | Location |
|---|---|---|---|
| Dye (DI) | 66°N, 53°E | American Samoa (AS) | 14°S, 170°W |
| Heimaey (HE) | 63°N, 20°W | New Caledonia (NC) | 21°S, 166°E |
| United Kingdom (UK) | 58°N, 6°W | Norfolk Island (NI) | 29°S, 168°E |
| Mace Head (MH) | 53°N, 10°W | Amsterdam Island (AI) | 38°S, 77°E |
| Crete Island (CI) | 35°N, 25°E | Cape Grim (CG) | 40°S, 144°E |
| Bermuda (BE) | 32°N, 65°W | Palmer (PA) | 65°S, 64°W |
| Tenerife (TE) | 28°N, 17°W | Dumont D'Urville (DU) | 66°S, 140°E |
| Midway Island (MD) | 28°N, 177°W | Mawson (MA) | 67°S, 63°E |
| Miami (MI) | 26°N, 80°W | Neumayer (NE) | 70°S, 8°W |
| Barbados (BA) | 13°N, 60°W | Halley Bay (HB) | 75°S, 26°W |
| Fanning Island (FI) | 4°N, 159°W | Kohnen (KO) | 75°S, 0°E |
| | | Dome C (DC) | 75°S, 123°E |

---

## Author Response (AR2)

We thank the referee and the editor for the helpful comments. Please find our responses below.

**1) Comment**
First of all, the authors have done a good job answering the comments. The responses are clear and addressed the issues.
Still, there is one thing I do not understand. The authors stated that they cannot find DMS values for the year 2007. However, in Müller et al. (2010), there are DMS values reported for May to June 2007 measured at the Cape Verde Atmospheric Observatory (CVAO). As mentioned in my former response, since 2006 DMS concentrations are permanently measured there (see Carpenter et al., 2010). This station is operated by the National Centre for Atmospheric Science, Department of Chemistry, University of York. As two of the Co-authors are working at this University and one also at the National Centre for Atmospheric Science, I do not understand why it is not possible to get data access. Are there lacks in the measurements for 2007? Please clarify this.
Overall, if this issue is clarified, the paper is suitable for publication in ACP.

**Response**: We have spoken to the P.I. of CVAO (Lucy Carpenter). Unfortunately, she informed us that the DMS data is not yet available to be shared as it still requires further processing and quality control.

**Changes in the manuscript**: No changes.

**2) Comment**
In addition, the expressions of multiphase or aqueous reactions somewhere in the manuscript are not very clear to me. Specifically, at Page 5, Line 24-28, are DMS, DMSO, MSIA and MSA gaseous or liquid species? Are their reactions with O3(aq) and OH(aq) the same as those aqueous-phase reactions shown in Table 1? The issue is that an index of (aq) is used for DMS, DMSO, MSIA and MSA in Table 1, but not in the text.

**Response**: The multiphase reaction starts with the uptake of gas-phase sulfur species into the cloud droplets and aerosols, followed by the reaction in the aqueous-phase of this sulfur species with the aqueous-phase oxidants $OH_{(aq)}$ and $O_{3(aq)}$. The aqueous-phase rate constant in Table 1 is for the aqueous part of the multiphase reaction. Therefore, in the updated manuscript, we denote multiphase reaction in the form of $X_{(g)}+OH_{(aq)}$ but rate constant for the aqueous-phase reaction in the form of $X_{(aq)}+OH_{(aq)}$.

**Changes in the manuscript**:
- Page 5, Line 24-28: change into:
For the multiphase reactions $DMS_{(g)} + O_{3(aq)}$, $DMSO_{(g)} + OH_{(aq)}$, $MSIA_{(g)} + OH_{(aq)}$, $MSIA_{(g)} + O_{3(aq)}$ and $MSA_{(g)} + OH_{(aq)}$ in cloud droplets and aerosols, we assume a first-order loss of the gas-phase sulfur species, following the parameterization described in Ammann et al. (2013) and Chen et al. (2017):

$$\frac{d[X_{(g)}]}{dt} = -\frac{c\gamma}{4} A[X_{(g)}] , \qquad\qquad (E4)$$

- Page 6, Line 12: add:
Gas-phase sulfur species taken up by aerosols and cloud droplets will be oxidized in the aqueous phase.

Change "$k_{X+Y}$ is the aqueous-phase reaction rate coefficient between X and Y ($M^{-1} s^{-1}$), as summarized in Table 1." into "$k_{X+Y}$ is the aqueous-phase reaction rate coefficient between aqueous-phase X and Y ($M^{-1} s^{-1}$), as summarized in Table 1."

- Page 4, Line 22-29: specify gas phase oxidation of DMS.

[revised manuscript text omitted]